# DeSUMOylation of a *Verticillium dahliae* enolase facilitates virulence by derepressing the expression of the effector *VdSCP8*

Xue-Ming Wu[1,2,4], Bo-Sen Zhang[1,2,4], Yun-Long Zhao[1,3,4], Hua-Wei Wu[1,2], Feng Gao[1,2], Jie Zhang [1,2], Jian-Hua Zhao [1,2] ✉ & Hui-Shan Guo [1,2] ✉

The soil-borne fungus *Verticillium dahliae*, the most notorious plant pathogen of the *Verticillium* genus, causes vascular wilts in a wide variety of economically important crops. The molecular mechanism of *V. dahliae* pathogenesis remains largely elusive. Here, we identify a small ubiquitin-like modifier (SUMO)-specific protease (VdUlpB) from *V. dahliae*, and find that VdUlpB facilitates *V. dahliae* virulence by deconjugating SUMO from *V. dahliae* enolase (VdEno). We identify five lysine residues (K96, K254, K259, K313 and K434) that mediate VdEno SUMOylation, and SUMOylated VdEno preferentially localized in nucleus where it functions as a transcription repressor to inhibit the expression of an effector *VdSCP8*. Importantly, VdUlpB mediates deSUMOylation of VdEno facilitates its cytoplasmic distribution, which allows it to function as a glycolytic enzyme. Our study reveals a sophisticated pathogenic mechanism of VdUlpB-mediated enolase deSUMOylation, which fortifies glycolytic pathway for growth and contributes to *V. dahliae* virulence through derepressing the expression of an effector.

*Verticillium dahliae* Kleb. is a soil-borne phytopathogenic fungus that causes vascular wilts in over 200 plant species worldwide[1,2]. *V. dahliae* can form microsclerotia, which function as a long-survival resting structure in the soil. Triggered by root exudates, microsclerotia germinate on the root surface and rapidly penetrate the root via the hyphopodium, which is an infectious structure that develops penetration peg for breaching host cells and secreting fungal effectors[3–7]. Several *V. dahliae* secretory effectors have been found to translocate into plant cells to modulate the host defense pathways[8–10]. A recent study has identified a substantial difference in the secretory effectors that prevent chitin-triggered host immunity between leaf-infecting and root-infecting plant pathogenic fungi. Leaf-infecting pathogens secrete chitin-binding proteins, which bind the fungal cell wall-derived chitin oligomers[11,12], to prevent chitin N-acetyl group from triggering host immunity[13,14]. Unlike leaf-infecting fungal pathogens, *V. dahliae*

and *Fusarium oxysporum*, another well characterized as a soil-borne fungal pathogen[15], secret a highly active enzyme polysaccharide deacetylase (PDA1), to directly deacetylate chitin oligomers and provide a subtle and efficient strategy to prevent chitin-triggered host immunity[16].

Recently, mounting evidence has shown that post-translational modifications also play critical roles for pathogen in overcoming host immune defenses. Post-translational modifications, including phosphorylation, methylation, acetylation, ubiquitination, and SUMOylation, regulate protein properties by covalently attaching regulatory groups to target amino acid residues[17–20]. SUMO is a small ubiquitin-like modifier expressed in all eukaryotes and is covalently attached to lysine residues of the target protein through multiple-step catalysis similar to ubiquitination[21,22]. Diverse SUMO proteins are found in different organisms. A single SUMO gene has been identified in yeast.

[1]State Key Laboratory of Plant Genomics, Institute of Microbiology, Chinese Academy of Sciences, Beijing 100101, China. [2]CAS Center for Excellence in Biotic Interactions, University of the Chinese Academy of Sciences, Beijing 100049, China. [3]Present address: Section of Cell and Developmental Biology, Division of Biological Sciences, University of California, San Diego, La Jolla, CA 92093, USA. [4]These authors contributed equally: Xue-Ming Wu, Bo-Sen Zhang, Yun-Long Zhao. ✉e-mail: zhao_jian_hua@hotmail.com; guohs@im.ac.cn

Four distinct SUMO proteins are encoded in the human genome. Of these, the mature forms of HuSUMO2 and HuSUMO3 are 97% identical, but share only 50% sequence identity with HuSUMO1[23–25]. HuSUMO4 is expressed in some special tissues, and whether it is processed to the mature form is unclear so far[24]. Among the six *Arabidopsis* SUMO proteins, AtSUMO1 and AtSUMO2 are highly related, and each shares 45 and 46% sequence identity with HuSUMO1[26]. SUMOylation is a reversible modification, which can be removed by ubiquitin-like protein-specific proteases (Ulp), such as Ulp1 and Ulp2 in yeast[27,28]. In mammals, there are six sentrin/SUMO-specific protease (SENP) homologs. SENP1-3 and SENP5 share similarity with Ulp1, whereas SENP6 and SENP7 closely relate to Ulp2[29,30]. Through comparative genomics and phylogenetic analysis, one ULP1-like and three ULP2-like SUMO protease subgroups have been identified in *Arabidopsis*[31].

In some plant and animal fungi, SUMOylation is required for both sexual and asexual development, cell cycle, stress responses, effector secretion and appressorium-mediated infection[18,32–35]. However, whether and how SUMOylation regulated the virulence of pathogenic fungi remains largely unclear, and neither of bona fide SUMO substrates in plant pathogenic fungi have been experimentally verified.

In this study, through a pathogenicity-deficient screening of T-DNA insertion mutant library of *V. dahliae*, we identify a low-virulence strain with a single copy of T-DNA integrated into a UlpB gene in *V. dahliae*, named VdUlpB. We find that VdUlpB is a deSUMOylase and mainly responsible for removing SUMO conjugations from substrate proteins in vivo. Further analysis shows that VdEno is a substrate of VdUlpB and five lysine residues in VdEno-mediated its SUMOylation. SUMOylated VdEno is more targeted to the fungal nucleus, binds to the promoter of a secretory effector gene *VdSCP8*, and represses its expression. In contrast, VdUlpB-mediated deSUMOylation of VdEno increases its cytoplasmic distribution, leading to derepressing *VdSCP8* expression and enhancing *V. dahliae* virulence. Overall, our study identifies VdEno as a bona fide SUMO substrate and its subcellular localizations are regulated by VdUlpB-mediated deSUMOylation, which is required for derepressing the transcription of *VdSCP8* and the virulence of *V. dahliae*.

## Results

### Identification of VdUlpB required for pathogenicity of *V. dahliae* in cotton plants

Through the pathogenicity-deficient screening of the T-DNA insertion mutant library of *V. dahliae*[36], we identified a low-virulence strain (Vd^T-DNA) with a single copy of T-DNA integrated in a small ubiquitin-like modifier (SUMO)-specific protease (Ulp) gene in *V. dahliae* (Supplementary Fig. 1a–d), named *VdUlpB*, by blasting the genome sequence of VdLs.17[37]. Phylogenetic analysis illustrated VdUlpB, together with human SENP6 and SENP7, belong to the yeast Ulp2 branch[29], while another homolog in *V. dahliae*, VdUlpA, is classified into a more divergent Ulp1 branch along with human SENP1/2 and SENP3/5 (Supplementary Fig. 1e).

To validate that the reduced virulence of T-DNA mutant was due to the disruption of VdUlpB, we generated a VdUlpB-knockout mutant (VdΔ*ulpb*) for further investigation. Similar to the T-DNA insertion mutant Vd^T-DNA, VdΔ*ulpb* showed reduced formation of melanin microsclerotia without altering hyphal growth (Fig. 1a, b). Of note, both Vd^T-DNA and VdΔ*ulpb* exhibited markedly reduced virulence in cotton plants, and no longer induced wilt symptoms in contrast to the wild-type (WT) *V. dahliae* V592 (Fig. 1c, d), indicating that VdUlpB is required for *V. dahliae* pathogenicity. We observed hyphae penetration and growth on medium for both WT V592 and VdΔ*ulpb* mutant strains (Supplementary Fig. 1 f). *V. dahliae* needs to overcome the reactive oxygen species (ROS) stress during host penetration[38]. To test whether VdUlpB is responsible for ROS tolerance, WT V592, Vd^T-DNA, and VdΔ*ulpb* mutants were grown on medium plates containing hydrogen peroxide (H_2O_2). No significant difference in hyphal growth

(colony diameter) between these strains (Supplementary Fig. 1g). Consistently, no significant difference in fungal biomass was detected in V592-infected and VdΔ*ulpb*-infected cotton plants at 5 days post inoculation (dpi) (Supplementary Fig. 1h). All these data indicated that the decrease pathogenicity of the VdΔ*ulpb* mutant was not due to failure in the initial colonization and fungal proliferation in cotton plants. Moreover, sequence alignment of human SENP7 and VdUlpB indicated that T-DNA was inserted into the conserved protease catalytic domain of VdUlpB (Supplementary Fig. 1i), causing Vd^T-DNA produced a truncated VdUlpB mRNA without the C-terminal sequence of the protease catalytic domain (Supplementary Fig. 1j). This suggests that the decreased melanin microsclerotia formation and pathogenicity of Vd^T-DNA were due to loss of SUMO-specific protease activity. To examine this hypothesis, VdΔ*ulpb* was compensated with either a full-length WT *VdUlpB* or a protease-deficient mutant *VdUlpBm*, which has a cysteine-to-serine mutation at residue 711 (C711S, Supplementary Fig. 1j) to eliminates the activity of SUMO-specific protease (see below Fig. 2a). Colony morphology and pathogenicity analysis showed that only WT *VdUlpB* but not *VdUlpBm* can restore VdΔ*ulpb* microsclerotia formation and pathogenicity in cotton plants (Fig. 1b–e). Taken together, our results demonstrate that *VdUlpB* is required for melanin microsclerotia formation and the pathogenicity of *V. dahliae*, which might depend on its activity that modifies SUMOylation.

### VdUlpB is a SUMO-specific protease

SUMO-specific protease is a dual-function enzyme that can catalyze SUMO precursor maturation prior to the conjugation and deconjugation of SUMO from SUMO-modified proteins. We thus examined whether VdUlpB was a protease responsible for the pre-SUMO processing and removal of SUMO modification. To this end, the SUMO homologous sequence was blasted in the VdLs.17 database using yeast SUMO protein sequence. Two SUMO homologs were found, but transcript was detected only in one locus and named VdSUMO (Supplementary Fig. 2a). We then purified the His-tagged catalytic domain of VdUlpB (His-VdUlpB^CD, 387–780 aa of VdUlpB), the His-tagged C711S mutant of VdUlpB^CD (His-VdUlpB^CDm), and the Strep-tagged VdSUMO precursor (Strep-pre-VdSUMO) (Supplementary Fig. 2b). In vitro co-incubation of His-VdUlpB^CD and Strep-pre-VdSUMO was conducted for 4 h, and VdSUMO with smaller molecular weight was detected in SDS-PAGE gel (Fig. 2a), indicating that VdUlpB^CD had efficiently cleaved the pre-VdSUMO to form mature VdSUMO. Notably, pretreatment with n-ethylmaleimide (NEM), a SUMO-specific protease inhibitor that alkylates catalytic cysteine, completely abolished VdUlpB^CD activity and left the pre-VdSUMO intact (Fig. 2a). Consistently, His-VdUlpB^CDm also failed to process pre-VdSUMO (Fig. 2a). This result indicates VdUlpB is capable to catalyze SUMO precursor maturation and this activity requires the catalytic cysteine.

We next examined the deconjugation activity of VdUlpB by co-incubating VdUlpB^CD and SUMO-modified human RanGAP1, which is well-known to be modified by SUMO. SUMOylated RanGAP1 was generated by conjugating human SUMO1 (HuSUMO1) onto GST-RanGAP1 using a commercial SUMOylation kit, and then was incubated with VdUlpB^CD. After incubation, HuSUMO1 was largely deconjugated from RanGAP1 by VdUlpB^CD, and the deconjugation of HuSUMO1 was diminished by C711S mutation and NEM pretreatment (Fig. 2b). Additionally, we repeated this assay by replacing HuSUMO1 with VdSUMO to confirm VdUlpB activity to deconjugate VdSUMO modification. As expected, the WT VdUlpB^CD but not the NEM-treated VdUlpB^CD nor the C711S mutant VdUlpB^CDm was capable to de-SUMOylate VdSUMO-modified RanGAP1 (Fig. 2b), indicating VdUlpB is a protease that can regulate SUMOylation of substrates. Moreover, both VdSUMO- and HuSUMO1-modified RanGAP1 can be detected by anti-human SUMO1 antibody (Fig. 2b), suggesting that there was a conserved epitope between HuSUMO1 and VdSUMO. We also observed a strong SUMO modification in the strain overexpressing Strep-VdSUMO with anti-

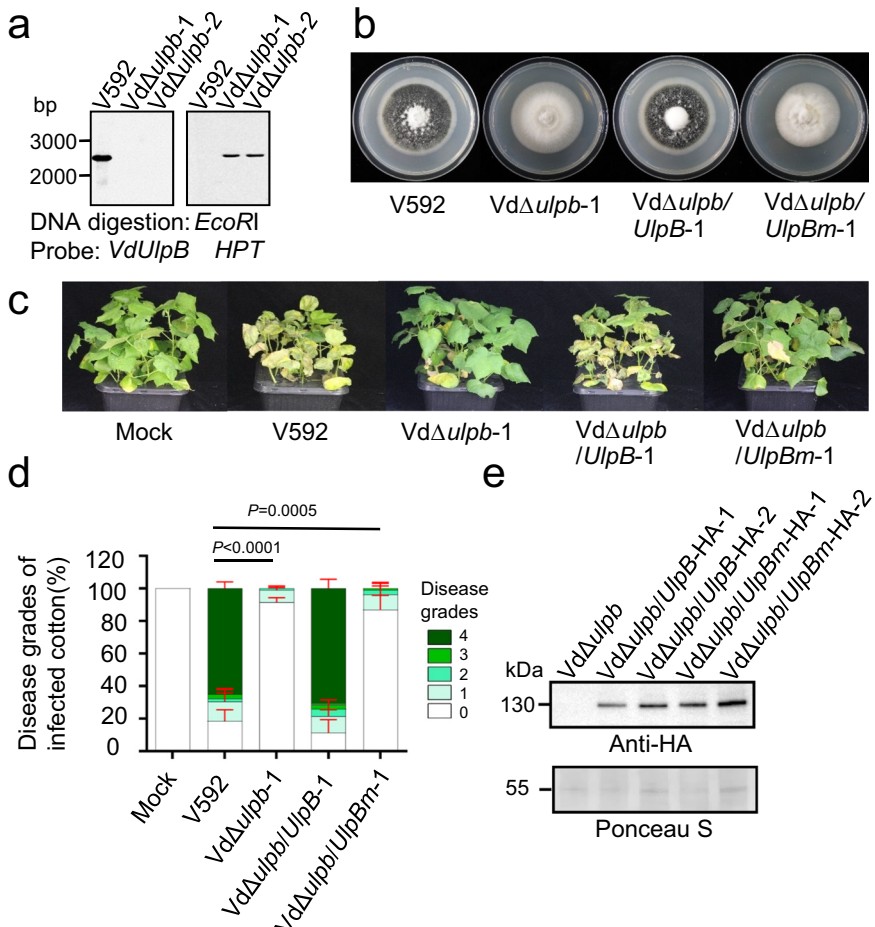

**Fig. 1 | *VdUlpB* is required for pathogenicity of *V. dahliae*. a** Confirmation of the knockout mutant VdΔ*ulpb* by Southern blotting. Genomic DNA was digested with *EcoR* I at 37 °C overnight. The probes were labeled with biotin. **b** Colony morphology of wild-type (WT) V592, *VdUlpB* knockout mutant VdΔ*ulpb* and complementary strains, VdΔ*ulpb*/*UlpB-1* and VdΔ*ulpb*/*UlpBm-1*, with WT *VdUlpB* or SUMO-specific protease site mutated *VdUlpBm*, on PDA plates after 19 days postincubation (dpi). **c** Disease symptoms of cotton plants infected with the indicated strains at 22 dpi. **d** The disease grades were divided into five levels of disease symptoms severity in cotton leaves. Disease grades were evaluated with three replicates of 36 plants for each inoculum (mean ± s.d., *t* test, two-sided). **e** Identification of complementation of VdΔ*ulpb* with WT *VdUlpB* (full-length of VdUlpB: 1063aa) or mutated *VdUlpB* (*VdUlpBm*) by western blotting with anti-HA antibody, and Ponceau staining served as a loading control. The experiments in **a**, **e** were repeated independently three times with similar results. Source data are provided as a Source Data file.

human-SUMO1 but not with anti-human-SUMO2/3 (Supplementary Fig. 2c). In contrast, a much weaker SUMOylation was detected in the SUMO ligase Ubc9 knockout mutant (VdΔ*ubc9*) (Supplementary Fig. 2d, e), revealing that anti-human SUMO1 antibody can detect both HuSUMO1 and VdSUMO. Collectively, these in vitro assays indicate that VdUlpB has the dual-functional activity of the SUMO-specific protease.

### VdUlpB mainly mediates substrate for deSUMOylation in *V. dahliae*

To validate the in vitro finding, we tested whether VdUlpB regulated SUMOylation of *V. dahliae* in vivo. For this purpose, whole proteins were extracted from *V. dahliae* for Western blotting with anti-human SUMO1 antibody. Compared to V592, either the *VdUlpB* mutation in Vd[T-DNA] or the depletion in VdΔ*ulpb* strongly enhanced the protein SUMOylation of *V. dahliae* (Fig. 2c, lanes 2 and 4), while complementation with the full-length WT *VdUlpB* reversed the increase in SUMOylation (Fig. 2c, lanes 3 and 5), supporting our in vitro finding that VdUlpB catalyzes SUMO deconjugation. We also generated a knockout mutant of *VdUlpA*. Unlike VdΔ*ulpb*, increased protein SUMOylation was not detected in VdΔ*ulpa* mutant compared to V592 (Fig. 2d). Combining in vitro and in vivo data, we conclude that VdUlpB is a deSUMOylase which is responsible for the protein deSUMOylation in *V. dahliae*.

### VdEno is a substrate of VdUlpB

VdUlpB regulates the global SUMOylation in *V. dahliae* (Fig. 2c). To identify SUMO-modified proteins and determine the potential SUMO-specific protease target, we first transformed a Strep-VdSUMO construct into VdΔ*ulpb* to create the VdΔ*ulpb*/Strep-SUMO strain for immunoprecipitation with anti-Strep. An anti-Strep-IPed sample was used for mass spectrometry (MS), and twenty-one potential proteins were obtained (Supplementary Table 1). Because SUMO modifications can change the molecular weight and/or the isoelectric point of target proteins, or cause degradation of target protein as well, we then compared the proteome of V592 and Vd[T-DNA] using two-dimensional gel electrophoresis to separate proteins on the basis of their size and of their charge. The differentially modified protein spots due to VdUlpB deficiency were quantified by using Image Master 2D Platinum software (Cytiva), the volume ratio between two strains higher than 2 (*p* value < 0.05) as a cutoff, a total of thirty-one proteins were identified by MS, including proteins involved in regulating fungal primary metabolism, biosynthesis, redox, and stress responses (Supplementary Table 2). Notably, the most evidently decreased protein in Vd[T-DNA] compared with V592 was an enolase homolog (Supplementary Fig. 3, arrow 9 red circle, named VdEno), which is also found in the anti-Strep-IPed proteins.

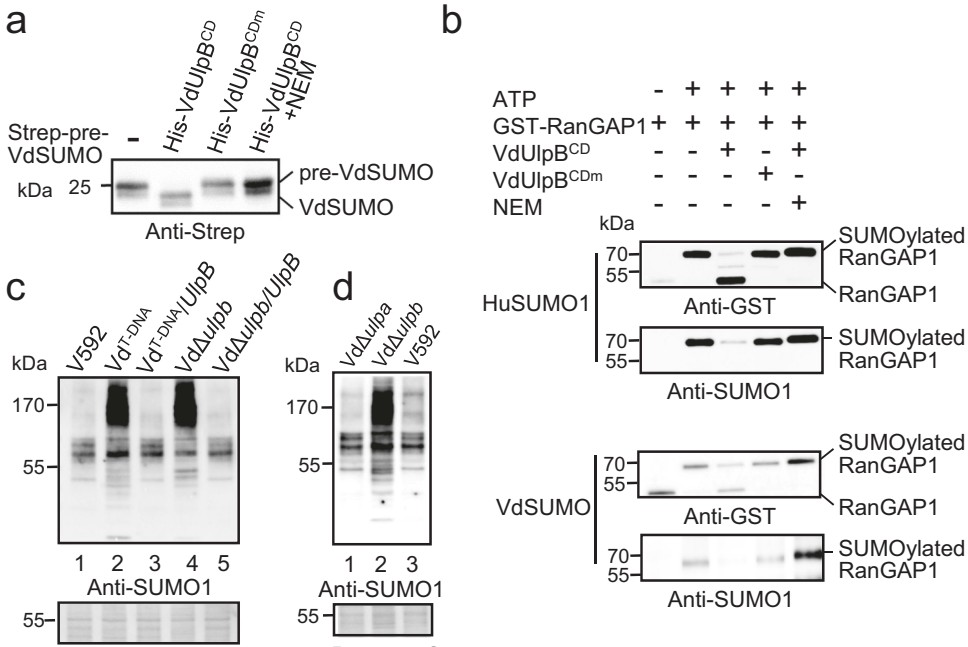

**Fig. 2 | VdUlpB is a SUMO protease and regulates SUMOylation of *V. dahliae*.**
**a** Examination of the SUMO precursor processing activity of VdUlpB[CD] in vitro. The precursor VdSUMO (Strep-pre-VdSUMO) was incubated with the VdUlpB catalytic domain (His-VdUlpB[CD]), mutated VdUlpB[CD] (His-VdUlpB[CDm]) or NEM-treated VdUlpB[CD] at room temperature for 4 h, followed by western blotting with anti-Strep antibody. **b** Examination of the SUMO isopeptidase activity of VdUlpB[CD] in vitro. Human SUMO1 (HuSUMO1) or VdSUMO was first attached to RanGAP1. Then, SUMOylated RanGAP1 was incubated with VdUlpB[CD], VdUlpB[CDm], or NEM-treated

VdUlpB[CD] at room temperature for 4 h. These products were analyzed by western blotting with anti-GST antibody and anti-human SUMO1 antibody. **c** Analysis of protein SUMOylation in vivo. Total proteins from the indicated strains were extracted and immunoblotted with anti-human SUMO1 antibody, and ponceau staining served as a loading control. **d** Analysis of protein SUMOylation with anti-human SUMO1 antibody in V592, VdΔ*ulpb*, and VdΔ*ulpa* strains. Ponceau staining served as a loading control. The experiments in **a**–**d** were repeated independently three times with similar results. Source data are provided as a Source Data file.

Enolase, which is first regarded as one of the most important enzymes in the glycolytic pathway[39,40], is a multifunctional protein according to its subcellular localization. However, the functions and subcellular localization of enolase related to its post-translational modifications is not well understood. Enolase was also reported to endow *Arabidopsis thaliana* and *Botrytis cinerea* with cold resistance[41,42]. We then tested cold stress response for *V. dahliae*, compared to WT V592, both Vd[T-DNA] and VdΔ*ulpb* revealed significantly dampened mycelial growth under cold stress (Supplementary Fig. 1 g). This result hints that *VdUlpB* deletion possibly affected the functions of VdEno, and prompted us to investigate whether VdEno could be a potential SUMOylation target and a substrate of VdUlpB.

To determine whether VdEno was a bona fide substrate of VdUlpB, we first examined the interaction between VdUlpB and VdEno. Due to the unsuccessful expression of the full-length *VdUlpB* in yeast or *Escherichia coli* for yeast-two-hybrid and Pull-down assays, we tried the TurboID-based proximity-labeling method that has been used in plant and animals for interactome analysis[43–45]. VdEno-TurboID was expressed in VdΔ*ulpb*/*UlpB*-HA or VdΔ*ulpb*/*UlpBm*-HA strains to produce VdΔ*ulpb*/*UlpB*-HA/Eno-TurboID or VdΔ*ulpb*/*UlpBm*-HA/Eno-TurboID strains (Fig. 3a). In the presence of biotin, we detected a special band corresponding to Eno-TurboID (Fig. 3b, red asterisk) and other signals (Fig. 3b), demonstrating that TurboID actively labeled biotin to VdEno and proximal proteins in *V. dahliae* (Fig. 3b). Immunoprecipitation of biotinylated proteins with streptavidin-coated magnetic beads, VdUlpB-HA but not VdUlpBm-HA was detected with anti-HA antibodies (Fig. 3c). This result indicates that VdUlpB interacted with VdEno which probably depend on its catalytic domain.

Then we assessed whether VdEno was SUMOylated in vitro. We purified Strep-tagged VdEno (Strep-VdEno) and the mature type of VdSUMO (Strep-VdSUMO) (Supplementary Fig. 2b and 4a). In the

presence of ATP, Strep-VdEno, and Strep-VdSUMO were co-incubated with human SUMO-activating enzyme E1 and SUMO-conjugating enzyme E2. After 2 h of incubation, reactions were stopped and subjected to western blotting with anti-Strep and anti-human SUMO1 antibodies. As shown in Fig. 4a, co-incubation caused VdEno to be modified by VdSUMO, indicated by slowly migrated bands between 70 and 130 kDa in western blotting (Fig. 4a). Notably, SUMO modification was specifically eliminated from VdEno by adding VdUlpB[CD] but not VdUlpB[CDm] (Fig. 4a). Furthermore, deSU-MOylation of VdEno was also detected by adding a full-length VdUlpB, but not a VdUlpBm, which were purified from V592-expressing HA-tagged-VdUlpB or HA-tagged-VdUlpBm (Supplementary Fig. 4b, c).

We next investigated whether VdEno was SUMOylated in vivo. To this end, VdEno with a Flag tag and VdSUMO with a Strep tag were expressed either alone or together in WT V592 to generate three stains of *V. dahliae*: VdEno-Flag (named VdEno strain), Strep-VdSUMO (VdSUMO strain), and VdEno-Flag/Strep-SUMO (VdEno/SUMO strain). VdEno-Flag was immunoprecipitated (IP) from *V. dahliae* using anti-Flag agarose beads and SUMOylation was blotted by anti-Strep antibody. Result showed that several enolase bands between 15 and 55 kDa were detected by anti-Flag antibody (Fig. 4b, lanes 2 and 3). Whether the multiple bands detected with anti-Flag were variants of VdEno, like the human enolase[46], or were degradation products requires further investigation. Nevertheless, slight signals of VdSUMO and SUMOylated VdEno were detected with anti-Strep antibody in the IP sample (Fig. 4b, lane 7), indicating that VdEno was modified with VdSUMO. Intense signals of SUMOylated VdEno were detected when the VdUlpB was deleted in the VdEno/SUMO strain (VdEno/SUMO/Δ*ulpb*) (Fig. 4b, lane 8), indicating that VdUlpB is essential for cleaving off the modified VdSUMO from VdEno.

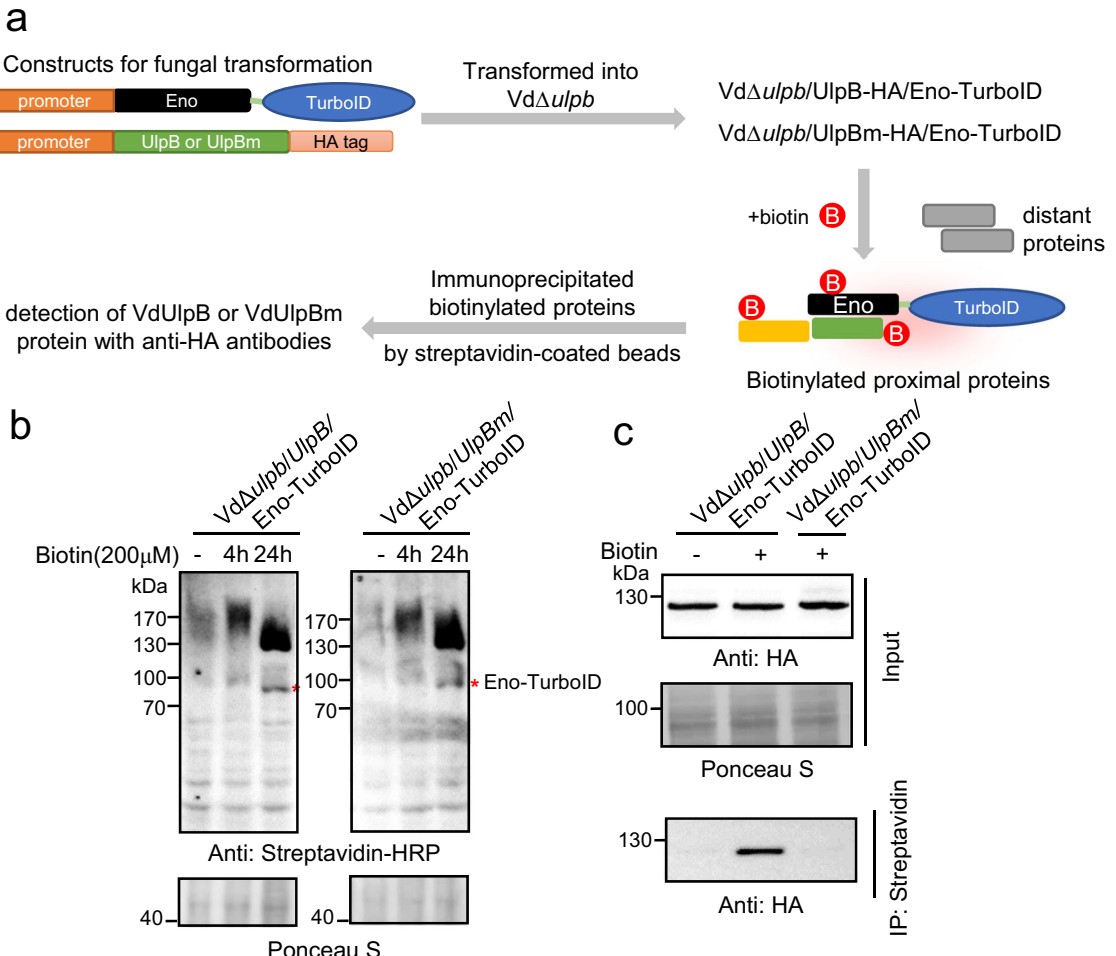

**Fig. 3 | VdUlpB interacted with VdEno in *V. dahliae*. a** The schematic diagram of TurboID-based proximity–labeling method. **b** Analysis of the biotinylated proteins by TurboID in *V. dahliae* treated with exogenous biotin after 4 h or 24 h. "-" indicates no treatment, and acts as negative control. Red asterisks indicate Eno-TurboID. **c** After treatment with exogenous biotin for 24 h, VdUlpB, but not VdUlpBm, was detected to interact with VdEno through immunoprecipitation with Streptavidin magnetic beads and analyzed by western blotting with anti-HA antibody. The experiments in **b**, **c** were repeated independently three times with similar results. Source data are provided as a Source Data file.

## Identification of SUMOylation sites in VdEno

To identify the SUMOylation sites in VdEno, VdEno and SUMOylated VdEno were immunoprecipitated using anti-Flag agarose beads from VdEno/SUMO/Δ*ulpb* strain, and separated by SDS gel. The proteins above 55 kDa, which were indicated by marker, were excised and subjected to MS. One putative SUMOylation site (K313) in VdEno was given by MS (Supplementary Fig. 5a). We substituted this lysine with arginine, and Strep-VdEno$^{K313R}$ was expressed in *E. coli* for in vitro SUMOylation assay (Supplementary Fig. 5b). No alteration in SUMOylation was detected for VdEno$^{K313R}$ mutation compared to Strep-VdEno (Supplementary Fig. 5c). We then used GPS-SUMO software (http://sumosp.biocuckoo.org/) to analyze VdEno protein sequence and four putative SUMOylation lysine residues (K96, K254, K259, and K434) were found. We substituted these four lysines with arginines individually or simultaneously, and Strep-tagged VdEno mutants (Strep-VdEno$^{K96R}$, Strep-VdEno$^{K254R}$, Strep-VdEno$^{K259R}$, Strep-VdEno$^{K434R}$, and Strep-VdEno$^{4K/4R}$) were expressed in *E. coli* for the in vitro SUMOylation assay (Supplementary Fig. 5b). Reduced SUMOylation signal was observed for VdEno with four K to R mutations (Strep-VdEno$^{4K/4R}$) but not the individual mutation (Supplementary Fig. 5c), suggesting that these four lysine residues are required for conjugating VdSUMO to VdEno in vitro. However, in vivo IP assay, VdEno SUMOylation signals were

only slightly inhibited in the VdEno$^{4K/4R}$/SUMO/Δ*ulpb* strain in which VdEno$^{4K/4R}$-Flag and Strep-*VdSUMO* were co-expressed (Supplementary Fig. 5d), suggesting the mutation of four predicted lysines are not sufficient to inhibit VdEno SUMOylation in vivo. We therefore generated a Strep-VdEno$^{5K/5R}$ (including K313) expressed in *E. coli* for the in vitro SUMOylation assay, and a VdEno$^{5K/5R}$/SUMO/Δ*ulpb* strain in which co-expressing VdEno$^{5K/5R}$-Flag and Strep-VdSUMO for in vivo SUMOylation assay. Clearly, reduced SUMOylation level was detected in vitro and in vivo (Fig. 4c, d), demonstrating that substituting the five lysine residues disrupts VdEno SUMOylation.

## VdUlpB regulates VdEno subcellular-localization

To understand how VdUlpB-mediated deSUMOylation regulates VdEno function, we examined the subcellular localization of both VdUlpB and VdEno in *V. dahliae*. We first observed the localization of VdUlpB. VdUlpB-GFP expressed in V592 localized in nucleus (Supplementary Fig. 6). In contrast to WT VdUlpB, VdUlpBΔN-GFP, lacking the N-terminal domain of VdUlpB, distributed in the cytoplasm (Supplementary Fig. 6). Of note, this mislocated VdUlpBΔN failed to restore the growth and pathogenicity defects of Vd$^{T-DNA}$ (Supplementary Fig. 1a–c), suggesting that the N-terminal domain is responsible for VdUlpB targeting to the nucleus, where is essential for SUMO protease function[47].

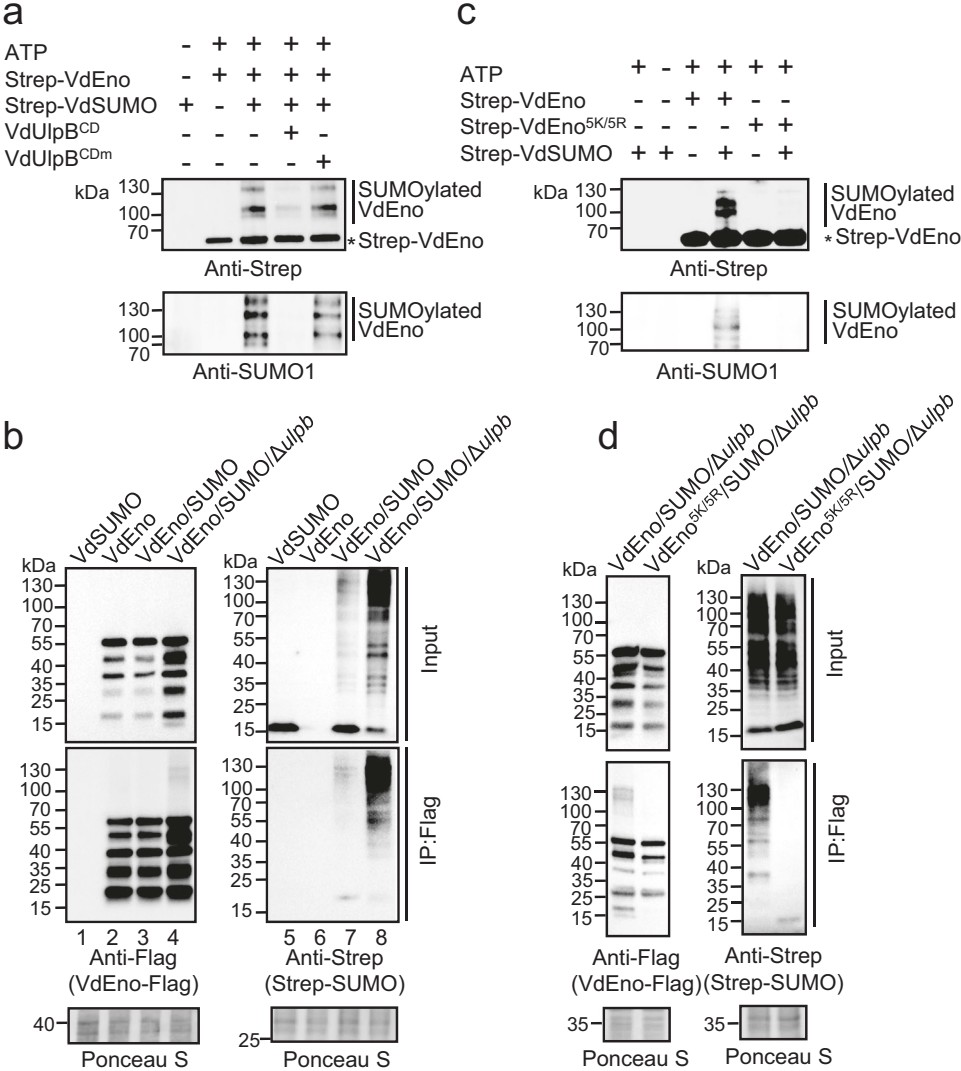

**Fig. 4 | VdUlpB regulates VdEno SUMOylation. a** VdEno is a SUMOylation substrate and deSUMOylated by VdUlpB. Co-incubation of Strep-Tagged VdEno and VdSUMO in the presence of E1, E2, and ATP at 37 °C for 2 h. The products were treated with VdUlpB[CD] or VdUlpB[CDm] at room temperature for 4 h followed by Western blotting with anti-Strep or anti-human SUMO1 antibody. **b** DeSUMOylation of VdEno by VdUlpB in vivo. The Strep-VdSUMO or *VdEno*-Flag constructs were expressed in WT V592, or co-expressed in the V592 and Vd*Δulpb* mutant. Total protein from these strains was immunoprecipitated with anti-Flag beads and immunoblotted with anti-Flag or anti-Strep antibody. Ponceau staining was used as a loading control. **c** Conformation of VdEno SUMOylation residues in vitro.

presence of E1, E2, and ATP, Strep-Tagged VdEno or VdEno[5K/5R] was co-incubated with VdSUMO at 37 °C for 2 h followed by western blotting with anti-Strep or anti-human SUMO1 antibody. **d** Conformation of VdEno SUMOylation residues in vivo. *VdEno*-Flag or *VdEno[5K/5R]*-Flag constructs were co-expressed with Strep-*VdSUMO* in the Vd*Δulpb* mutant. Total protein from these samples was immunoprecipitated with anti-Flag beads and immunoblotted with anti-Flag or anti-Strep antibody. Ponceau staining was used as a loading control. The experiments in **a**–**d** were repeated independently three times with similar results. Source data are provided as a Source Data file.

Enolase has been observed in multiple subcellular localizations to perform various functions[48]. We, therefore, tested if VdEno also localized in the nucleus and if the localization of VdEno was affected by VdUlpB-mediated deSUMOylation. To do so, a VdEno-GFP construct was transformed into the WT V592 and mutant Vd*Δulpb*. In the WT V592 strain, VdEno-GFP was observed in both the nucleus and the cytoplasm, while in the Vd*Δulpb* strain, VdEno-GFP was more centralized to the nucleus (Fig. 5a), indicating SUMOylation promotes VdEno nuclear localization. In support of this, VdEno[5K/5R]-GFP, which losses SUMOylation sites, is more favorably localized in the cytoplasm (Fig. 5a).

We also separated cytoplasmic and nuclear sections from both VdEno/SUMO and VdEno/SUMO/*Δulpb* strains, and intense SUMOylated VdEno signals were detected with anti-Strep antibody in the nuclear IP sample of the VdEno/SUMO/*Δulpb* strain, and they had more

pronounced higher molecular weight bands than the VdEno/SUMO strain (Supplementary Fig. 7, lane 8). Taken together, our results demonstrate that VdEno is localized in both the nucleus and the cytoplasm, and SUMOylation of VdEno increases its targeting to the nucleus.

**Cytoplasmic localized VdEno functions as a glycolytic enzyme**

Enolase was first regarded as one of the most important enzymes in the glycolytic pathway[39,40], which can catalyze the conversion of 2-phosphoglycerate to phosphoenolpyruvate[49]. We then investigated the enzyme activity of VdEno. The cytoplasmic and nuclear sections from the VdEno-GFP strain were separated, and VdEno-GFP from the cytoplasmic and nuclear total proteins was immunoprecipitated with anti-GFP agarose beads. Enolase glycolytic activity was tested using an enolase activity colorimetric assay kit by detecting the intermediate

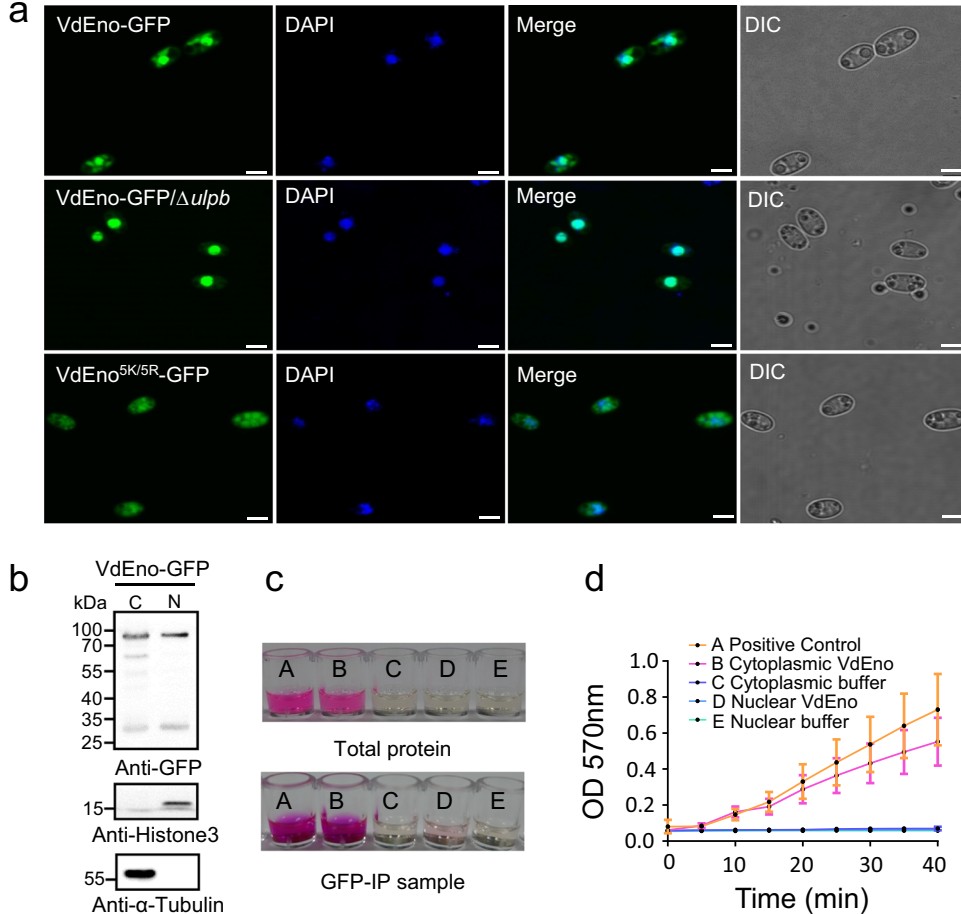

**Fig. 5 | VdUlpB regulates VdEno sub-localization and cytoplasmic VdEno functions as a glycolytic enzyme. a** SUMOylation of VdEno increases its targeting to the nucleus. VdEno localization was observed in VdEno-GFP, VdEno-GFP/Δ*ulpb*, and VdEno^5K/5R-GFP strains under confocal laser scanning microscopy (CLSM). Scale bar = 3 μm. The experiments were repeated independently three times with similar results. **b** Cytoplasmic and nuclear VdEno were separated from the VdEno-GFP strain (samples B and D in **c**, **d**) and immunoprecipitated with anti-GFP. Anti-histone 3 and anti-α-tubulin antibody, the nuclear and cytoplasmic markers, respectively, were used as indications. The experiments were repeated independently three times with similar results. **c** Enolase activity was detected by using an enolase activity colorimetric assay kit with positive (A) and buffer controls (C, E). **d** Enolase activity in cytoplasmic and nuclear VdEno (mean ± s.d., *n* = 3 biologically independent samples). Source data are provided as a Source Data file.

product of the enolase catalyzed reaction[50]. Enolase glycolytic activity was observed in the total proteins and VdEno-GFP IP samples from cytoplasmic sections but not from the nuclear sections (Fig. 5b–d), indicating nucleus-localized VdEno does not function as a glycolytic enzyme and VdUlpB-mediated deSUMOylation of VdEno increases its cytoplasmic distribution to function as a glycolytic enzyme. The deletion of VdEno was not successful, possibly due to the lethality in the absence of a glycolytic enzyme in *V. dahliae*. Taken together, our data demonstrate the important role of VdUlpB in deSUMOylation of VdEno for the cytoplasmic glycolytic pathway.

**Nuclear-localized VdEno functions as a transcription repressor**
Knocking out for VdUlpB markedly reduced *V. dahliae* virulence, we, therefore, investigated whether the nuclear non-glycolytic VdEno acts as a transcription factor and regulates the expression of potential virulence genes. VdEno was aligned to the human and *Arabidopsis* enolase homologs, showing that VdEno also contains DNA binding domain and repression domain (Supplementary Fig. 8a). The human c-myc DNA promoter element "TCGCGCTGAGTATAAAAGCCGGTTT" bound by enolase[41,42] was used to blast the genome sequence of VdLs.17 to search putative VdEno targets. Putative hits in promoters of genes including those known encoding secretory proteins (SCPs), *VdSCP1* to *VdSCP10* and *VdSCP41*[7–9,51], were found (Supplementary

Fig. 9a). The expression levels of *SCP* genes were first examined in WT V592 and Vd*Δulpb* mutant. Significantly reduced expression of *VdSCP8* was detected in Vd*Δulpb* mutant compared to that in WT V592, while no obvious alteration was found for other *SCP* genes (Supplementary Fig. 9b).

We, therefore, performed electrophoretic mobility shift assays (EMSAs) to examine whether VdEno could bind directly to the *VdSCP8* promoter. The conserved DNA binding domain of VdEno (VdEno^BD) was purified using *E. coli* (Supplementary Fig. 8b) and incubated with a 100 bp end-labeled putative enolase binding sequence from the *VdSCP8* promoter DNA containing the "c-myc promoter element" (Supplementary Fig. 9c). A slower migrating DNA-protein complex was detected in the presence of VdEno and labeled *VdSCP8* DNA (Fig. 6a), and this binding complex was clearly abolished by adding a 50-fold molar excess of unlabeled *VdSCP8* promoter DNA, but not unrelated DNA without enolase binding site (Fig. 6a), demonstrating the direct binding between VdEno and the *VdSCP8* promoter. Consistently, we also confirmed their binding in vivo using ChIP-qPCR assay (Fig. 6b). The promoter region of *VdSCP8* was significantly amplified in the VdEno-GFP pull-down sample, but not in IgG control, verifying that VdEno binds *VdSCP8* promoter in vivo (Fig. 6b). Notably, in *VdUlpB* depletion strain, much more *VdSCP8* promotors were enriched in VdEno-GFP pull-down sample than WT, while *VdUlpB* complementation reduced the enrichment of *VdSCP8*

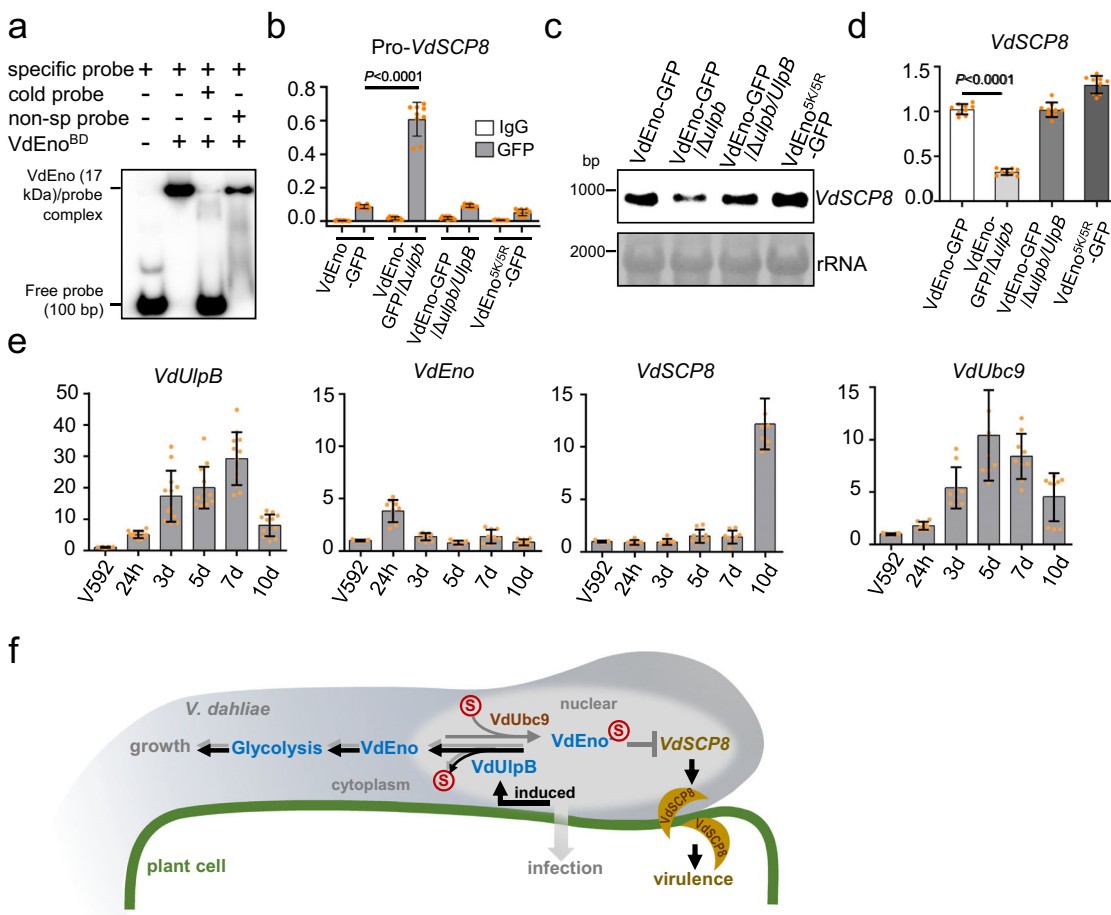

**Fig. 6 | VdEno in the nucleus functions as a transcriptional repressor of a *V. dahliae* effector. a** Specific DNA binding assay of VdEno in vitro. The binding domain of VdEno (VdEno$^{BD}$) was incubated with a specific probe cloned from the *VdSCP8* promoter region (described in Supplementary Figure 9c), cold probe, or the nonspecific probe followed by EMSA analysis. **b** Specific DNA binding assay of VdEno in vivo. VdEno was immunoprecipitated with anti-GFP antibody and IgG acts as negative control, and analyzed its binding capacity to the *VdSCP8* promoter region in the indicated strains by ChIP-qPCRmean ± s.d., $n = 9$ individual data points are shown, t-test, two-sided). **c, d** Detection of *VdSCP8* expression. *VdSCP8* expression in the indicated strains was examined by northern blotting (**c**) and RT-qPCR (**d**) (mean ± s.d., $n = 9$ individual data points are shown, *t* test, two-sided). **e** Transcript patterns of *VdUlpB, VdEno, VdSCP8,* and *VdUbc9* during infection (mean ± s.d., $n = 9$ individual data points are shown, one-way ANOVA followed by Tukey's multiple comparisons test). Total RNA was extracted from the cotton roots infected with V592 at different time points. h: hours, d: days. *V. dahliae elf1*-α was used as an internal control. The levels of the genes from V592 cultured in vitro relative to *elf1*-α was arbitrarily designated as 1. **f** A proposed working model for the role of *VdEno* in regulating the growth and pathogenicity of *V. dahliae*. The nuclear-localized VdEno is attached to VdSUMO (S) by E2 ligase VdUbc9 and deSUMOylated by the deSUMOylase VdUlpB (gray arrows). In nucleus, VdEno functions as a transcriptional repressor without glycolytic enzyme activity that binds to the promoter of the secretory effector gene *VdSCP8* (gray lines). VdUlpB-mediated deSUMOylation of VdEno promotes its cytoplasmic distribution for glycolysis-functioning for fungal growth and cold stress response (gray arrows); Increased *VdUlpB* during cotton plant infection (black bold arrows) and this leads to derepressing *VdSCP8* transcription and increasing VdSCP8 that translocated from *V. dahliae* to the plant cells and facilitates virulence independent of the glycolytic enzyme activity of VdEno. Source data are provided as a Source Data file.

promotor to a level comparable to WT (Fig. 6b). Faint amplification of *VdSCP8* promotors was shown in VdEno$^{5K/5R}$-GFP pull-down sample (Fig. 6b), consistent with its almost cytoplasmic distribution (Fig. 5a). Because of the in vitro binding assay using *E. coli*-produced VdEno (Fig. 6a), we reason that SUMOylation is not required for VdEno binding to DNA but restricts its nuclear localization (Fig. 5a) leading to advantaging its binding with *VdSCP8* promoter.

To further confirm whether VdEno acts as a transcriptional repressor on *VdSCP8*, *Agrobacterium* spp.-mediated luciferase (Luc) expression in plants was conducted. Transient expression of VdEno greatly repressed expression of *VdSCP8* promoter-derived Luc gene (Pro$_{(SCP8)}$-Luc) but not when the "c-myc promoter element" hit region was deleted in the *VdSCP8* promoter (Pro$_{(SCP8mut)}$-Luc) (Supplementary Fig. 10a−c). Moreover, VdEno-repressed transcription was not detected on other tested *SCP* promoters (Supplementary Fig. 10d), suggesting VdEno specifically represses *VdSCP8* transcription.

We further investigated the effect of VdEno on *VdSCP8* transcription in vivo. *VdSCP8* mRNA was examined using Northern blotting in VdEno-GFP wild-type strain, VdEno-GFP/Δ*ulpb* mutant strain, VdEno-GFP/Δ*ulpb*/UlpB complementary strain, and VdEno$^{5K/5R}$-GFP strain. We found that *VdSCP8* expression was decreased in VdEno-GFP/Δ*ulpb* mutant compared to WT, complementary, and VdEno$^{5K/5R}$-GFP strains (Fig. 6c), suggesting VdEno SUMO modification facilitates its inhibition in *VdSCP8* expression. Reverse transcription with quantitative real-time PCR (RT-qPCR) obtained similar results (Fig. 6d). Furthermore, we created a complementary strain VdΔ*scp8*/*SCP8*$^{Pmut}$, in which the VdEno binding site in the promoter of *VdSCP8* was deleted. *VdSCP8* expression was increased in VdΔ*scp8*/*SCP8*$^{Pmut}$ compared to VdΔ*scp8*/*SCP8* (Supplementary Fig. 11c), in agreement with ridding of VdEno-repressed transcription. Taken together, our data demonstrated that SUMOylation of VdEno increases its nuclear targeting which in turn

enhances VdEno transcriptional repressor action, and VdUlpB is essential to restore *VdSCP8* expression by deSUMOylating VdEno.

### VdSCP8 is an effector and regulated by VdUlpB/VdEno module

VdSCP8 is a previously identified secretory protein that translocates from *V. dahliae* to plant cytoplasmic streams and aggregates along the peripheral regions[7,8]. However, whether VdSCP8 had an effect on *V. dahliae* pathogenicity is not known. We generated knockout mutants of *VdSCP8* (VdΔ*scp8*) and complementary strains (VdΔ*scp8/SCP8*). VdΔ*scp8* displayed alternate growth of melanized microsclerotia and mycelia but exhibited reduced virulence in cotton plants compared to WT V592, and VdΔ*scp8/SCP8* restored its morphology and virulence to V592 (Supplementary Fig. 11a, b). Moreover, VdΔ*scp8/SCP8^Pmut* infection caused more severe wilting symptoms compared with VdΔ*scp8/SCP8* infection (Supplementary Fig. 11d). The increased virulence of VdΔ*scp8/SCP8^Pmut* correlated with VdEno-regulated *VdSCP8* expression. We cannot rule out that deletion of the VdEno binding site in *VdSCP* promoter might also impair interaction of another regulator, or alteration of the gene structure. Nevertheless, our data indicate that VdSCP8 is a secretory effector contributing to *V. dahliae* virulence.

Finally, we examined the expression patterns of *VdUlpB*, *VdEno* and *VdSCP8* during cotton root infection. To this end, RT-qPCR analysis was performed with the total RNA extracted from the infected cotton roots. RT-qPCR results showed that the transcript level of *VdUlpB* was first induced at 24 h post inoculation (hpi) and continued to increase to 7 dpi (Fig. 6e). The expression level of VdEno was slightly induced at 24 hpi but returned to basal level during other infection time points (Fig. 6e). The induced expression of *VdSCP8* was detected from 5 dpi, and by 10 dpi there was a prominent induction (Fig. 6e). This result was consistent with the period of leaf wilt visually apparent and supported the idea that *VdSCP8* plays a role in virulence of *V. dahliae* during cotton plant infection. In viewing the lowered expression of *VdUlpB* after 7 dpi (Fig. 6e), we found that the expression of the E2 conjugating enzyme coding gene *VdUbc9*, after its induction at 24 hpi, also began to decrease at 7 dpi (Fig. 6e), exhibiting a similar expression pattern as *VdUlpB* during the cotton plant infection. However, lower induction level of *VdUbc9* was detected compared with that of *VdUlpB*, especially at 7 dpi (Fig. 6e), suggesting VdUlpB-mediated deSUMOylation was the dominant process during plant infection, which in turn promoted *VdSCP8* expression. Taken together, our data suggest that SUMOylation of VdEno was essential to limit the *VdSCP8* at a basic level in in vitro growth. VdUlpB-mediated deSUMOylation of VdEno could be important in regulating *VdSCP8* expression during early plant infection, while at the later infection time points, reduction in both *VdUbc9* and *VdUlpB* resulted in reduced post-translational SUMOylation, thus no need of deSUMOylation of VdEno and more distribution in the cytoplasm, which allowed it to function as a glycolytic enzyme, and this led to derepression of *VdSCP8* in the nucleus.

## Discussion

Here, we uncovered a virulence strategy of a soil-borne pathogenic fungus by studying the SUMO-specific protease-mediated deSUMOylation. We found that VdUlpB is mainly responsible for removing the SUMO conjugation from substrate proteins and *V. dahliae* uses VdUlpB to de-SUMOylate transcription repressor VdEno, which in turn derepresses an effector gene *VdSCP8* to promote plant infection, meanwhile, deSUMOylated VdEno facilitates its cytoplasmic distribution to function as a glycolytic enzyme which essential for the fungal growth (Fig. 6f).

SUMO-mediated post-translational modification is a highly dynamic process and renders the plasticity to target protein functions, which is essential to achieve the coordinated regulation of multiple cell activities. Within this process, SUMO-specific protease plays a central role in deconjugating SUMO from SUMO-modified proteins and

catalyzing the maturation of SUMO precursor, which is important for SUMOylation homeostasis[28,52]. *V. dahliae* contains two SUMO-specific proteases, VdUlpA and VdUlpB. Our data indicate that VdUlpB which belongs to the yeast Ulp2 branch is a deSUMOylase for catalyzing SUMO deconjugation from modified proteins, while VdUlpA which is classified into the more divergent Ulp1 branch (Supplementary Fig. 1e) possibly carry out SUMO precursor maturation as previously reported in *Saccharomyces cerevisiae*[28].

By characterizing the VdUlpB T-DNA insertion mutant using proteomic approach, we identified a small portion of *V. dahliae* endogenous proteins that could be regulated by SUMOylation and verified that VdEno is a substrate of VdUlpB. Enolase, which was first described as a key glycolytic enzyme, is a multifunctional protein with the ability to function as plasminogen receptor on the cell surface and bind DNA as transcription repressor[53,54]. While the full-length enolase serves as glycolytic enzyme, interestingly, a shorter alternative translation product of enolase, *c-myc* binding protein (MBP-1), has been shown to negatively regulates *c-myc* transcription by binding to the P2 promoter in animal[55–58]. As a transcription factor, MBP-1 has suppressor activities on different types of tumors. Overexpression of MBP-1 inhibits the proliferation, migration, and invasion of cancer cells by regulating gene expression[48]. The small molecule ENOblock, which inhibits enolase activity and induces MBP-1 nuclear localization, decreased the migration and invasion of human colon carcinoma cells[59,60]. The transcription repressor activity of a truncated enolase was also identified in plants and fungus[41,42,61,62]. In *Arabidopsis*, MBP-1-like protein has little enolase activity in nuclei, and its accumulation is limited by ubiquitin-dependent destabilization[61,63]. Tandem mass spectrometry data showed that the *Saccharomyces cerevisiae* enolase is SUMOylated. Of note, enolase was reported to have several post translational modifications, such as phosphorylation, ubiquitylation, and SUMOylation in different species[64–67]. However, how post-translational modifications of enolase regulate its distinct functions and subcellular localizations is not well understood. In this study, we identified VdEno SUMOylation and found five lysine residues in VdEno essential for SUMOylation. Importantly, SUMOylated VdEno preferentially localizes in the nucleus, consistent with that SUMOylation mainly happens in nucleus[68,69]. We further found that increased nuclear localization of SUMOylated VdEno in *VdUlpB* knockout mutant represses transcription of *VdSCP8* (Figs. 5, 6c, and 6d). Whether the multiple bands detected with anti-Flag (Fig. 4b) were degradation products or included a truncated VdEno like human MBP-1, requires further investigation. Our data, however, raise the alternative possibility that SUMOylation might affect MBP-1 tumor suppressor activity by altering its subcellular localization.

Infection strategies are intricate for root pathogens that are successful underground, and as the most notorious plant pathogen of the *Verticillium* genus, *V. dahliae* must possess a variety of virulence regulatory techniques during plant infection. Number of secretory effectors of *V. dahliae* have been detected to be induced during plant infection to counter host immune defenses[7–9,16]. Of note, the *VdSCP8* knockout mutant displays significantly reduced wilt symptoms in cotton plants compared to the WT V592, indicating that VdSCP8 is an effector contributing to *V. dahliae* virulence. Our data demonstrate that VdUlpB-mediated deSUMOylation of VdEno is important for *V. dahliae* pathogenicity through controlling expression of the effector VdSCP8, and provide for the first time the regulation mode for an effector gene being repressed in vitro and induced during plant infection.

In addition to the derepression of *VdSCP8* transcription, VdUlpB-mediated deSUMOylation of VdEno increased its cytoplasmic distribution, which probably enhances VdEno-mediated glycolysis activity in cytosol. The deletion of VdEno was not successful, indicating that VdEno's glycolysis activity is essential for *V. dahliae* development. Together with that the *VdUlpB* knockout mutant reduced melanin

production, our data suggest that VdUlpB-mediated VdEno deSU-MOylation plays an important role in orchestrating VdEno glycolytic ability and transcription repressor function to promote *V. dahliae* growth and infection in plants. As a transcriptional factor, we can not rule out that VdEno probably regulate a variety of genes which also play roles in the fungal growth and virulence. The cytoplasmic lysates from WT and VdΔ*ulpb* mutant (VdEno-GFP and VdEno-GFP/Δ*ulpb*) have a comparable glycolytic activity (Supplementary Fig. 12), suggesting that the transcription factor activity of enolase, instead of the glycolytic activity, contributes to *V. dahliae* infection. Being an essential gene, SUMOylation regulation of VdEno should be vital to regulate its distinct subcellular localizations to undertake different functions in *V. dahliae* life cycle. Previous studies also found that, in *Candida glabrata*, the pathogen causing candidemia in human, CgUlp2 deletion led to impaired growth, and reduced its colonization of specific tissues in host[70]. Among *Aspergillus* spp., such as *A. flavus* and *A. nidulans*, loss of UlpA or UlpB resulted in sexual and/or asexual developmental defects. Furthermore, we found that deletion of VdUlpB in a tomato isolate *V. dahliae* strain JR2 (VdΔ*ulpb*[JR2]) showed no obviously developmental defect. However, VdΔ*ulpb*[JR2] strain showed significantly reduced degree of stunting of the plants on tomato when compared with WT JR2 (Supplementary Fig. 13). Taken together, all these results support the importance of SUMO-specific proteases in the entire life cycle of pathogenic fungi.

Recent study has shown that the *V. dahliae* ISW2 chromatin remodeling complex plays roles in positioning nucleosomes and gene expression in response to reactive oxygen species stress during development and plant infection[38]. In this study, we further show that post-translational SUMO/deSUMOylation play essential roles in regulation of *V. dahliae* virulence gene expression during plant infection, which could be necessary for successful infection of *V. dahliae*. In line with this, transcription regulation-related proteins are the most abundant targets of SUMOylation identified in *A. nidulans*[18,69], as well as in other organisms[71]. Intriguingly, the aforementioned ISW2 chromatin remodeling complex is also SUMOylated in *A. nidulans*[18,69]. Therefore, ISW2 complex could be a promising substrate of VdUlpB, which is required for plant infection, and worth further investigation in the future. Other putative VdUlpB substrates, such as those involved in regulating primary metabolism and biosynthesis (Supplementary Fig. 3 and Supplementary Table 2) are also worthy of further investigation. Undoubtedly, the growth and pathogenicity defects of the VdUlpB knockout mutant was not attributed only to the dysregulation of VdUlpB on VdEno. Nevertheless, our data provide evidence that VdUlpB/VdEno and VdEno/VdSCP8 regulatory modules play important roles in *V. dahliae* growth and pathogenicity.

## Methods

### Identification of T-DNA insertion mutant
To determine the copies of T-DNA in this mutant, genomic DNA was digested with *EcoR* I, *BamH* I, and *Xba* I, separated, and transferred to the HybondTM-N⁺ membrane (GE, RPN303B). The probe was amplified with the primer pair EGFPprobe-F/R (Supplementary data 1) and labeled with $^{32}$P using the Random Prime Labeling System Redi Prime II (GE Healthcare). Thermal asymmetric interlaced PCR (TAIL-PCR) was performed to identify the insertion site in Vd[T-DNA] as previously described[36]. To identify the *VdUlpB* (VDAG_01376) expression pattern in Vd[T-DNA], RNA was separated and transferred to the HybondTM-N⁺ membrane. The probe was amplified with the primer pairs UlpBprobe-F/R (Supplementary data 1) and labeled with $^{32}$P as described above.

### Penetration and pathogenicity assays
Hyphal penetration assays[7] and infection experiments[36] were performed as previously described in our laboratory. Disease severity was determined by counting the wilting symptom degrees on the upland cotton leaves. The disease grades were divided into five levels of severity of disease symptoms on cotton leaves (0, no visible wilting or yellowing symptoms; 1, one or two cotyledons wilted or dropped off; 2 and 3, one or two true leaves wilted or dropped off; and 4, all leaves dropped off or the whole plant has died). Tomato infection assays were followed by a previous description[72]. The infection assays were repeated at least three times to confirm reproducibility.

### Detection of *VdSUMO*
Based on a BLASTP search using yeast SUMO protein sequence, two SUMO homologs (VDAG_02409 and chromosome 6, 1626548–1626908 bp) from V592 were found. To conform their transcripts, specific reverse transcription PCR was conducted (Vazyme, R233-01). Primers are listed in Supplementary Data 1.

### Identification of VdUlpB catalytic activity
To detect the processing activity of VdUlpB, Strep-pre-VdSUMO was incubated with VdUlpB[CD], VdUlpB[CDm], or NEM-treated VdUlpB[CD] for 4 h at room temperature. Plasmid construction and transformation were described in Supplemental Methods. SDS-PAGE was analyzed by immunoblotting with a 1:5000 dilution of anti-Strep antibody (Easybio, BE2038). To examine the deconjugation activity of VdUlpB, SUMOylated RanGAP1 was co-incubated with VdUlpB[CD], VdUlpB[CDm], or NEM-treated VdUlpB[CD] for 4 h at room temperature. SDS-PAGE was analyzed by immunoblotting with a 1:5000 dilution of anti-GST (CWBIO, CW0144) and a 1:1000 dilution of anti-human SUMO1 (ENZO, BML-PW9460) antibody.

### TurboID-based proximity-labeling
To generate the expressing plasmid pSul-VdEno-TurboID construct, VdEno, and TurboID sequences were cloned. The corresponding fragments were ligated into the *BamH* I/*EcoR* I-linearized pSul binary vector. The primers used above are listed in Supplementary Data 1. pSul-VdEno-TurboID construct was transformed into VdΔ*ulpb*/*UlpB* or VdΔ*ulpb*/*UlpBm* strains, to get VdΔ*ulpb*/*UlpB*-HA/Eno-TurboID or VdΔ*ulpb*/*UlpBm*-HA/Eno-TurboID strains.

The strains were cultured in liquid Czapek-Dox medium for 2 days, and treated with 200 µM biotin for 4 h or 24 h, followed by washing with PBS for twice. For total proteins, cultures were lysed with extraction buffer (10 mM Tris pH 8.0, 150 mM NaCl, 0.5 mM EDTA, 1% Triton X-100, 1×Protease inhibitor). Biotinylated proteins were immunoprecipitated with streptavidin-coated magnetic beads and analyzed by Western blotting with anti-HA antibodies to detect VdUlpB or VdUlpBm protein.

### In vitro VdEno SUMOylation and deSUMOylation assays
SUMOylation experiments in vitro were carried out with a SUMOylation Kit (ENZO, BML-UW8955-0001). To determine whether the catalytic domain of VdUlpB could cleave SUMO from SUMOylated VdEno, VdUlpB[CD] or VdUlpB[CDm] were incubated with the SUMOylated VdEno. To examine the full-length VdUlpB activity, VdUlpB or VdUlpBm proteins were immunoprecipitated from V592/VdUlpB-HA or V592/VdUlpBm-HA strain, and incubated with the SUMOylated VdEno as described above.

### Identification of the SUMOylation sites in VdEno
To identify the SUMOylation sites in VdEno, mass spectrometry was performed. Briefly, SUMOylated VdEno proteins, immunoprecipitated with anti-Flag agarose beads (Sigma, A2220) from the VdEno/SUMO/Δ*ulpb* strain (n = 1), were separated by SDS-PAGE gels. The excised gels above 55 kDa were reductively alkylated with IAA, followed by dehydration with 100% acetonitrile. The gel pieces were digested with trypsin at 37 °C for 16 h and then with chymotrypsin for 4 h. The digestion solution was collected in a new tube. The peptide mixture was desalinated and dried with SpeedVac, and then resuspended in 0.1% TFA for MS analysis.

The peptide sample was analyzed on Thermo Fisher LTQ Orbitrap ETD mass spectrometry coupled to a Thermo Fisher Easy-nLC 1000 by AIMSMASS Co., Ltd. (Shanghai, China). Solvent A (0.1% formic acid, v/v) and solvent B (100% acetonitrile) were used to separate the peptides. The flow rate and gradient length were set as follows: 0 min, 96% A/4% B, 300 nL/min; 2 min, 93% A/7% B, 300 nL/min; 30 min, 85% A/15% B, 300 nL/min; 55 min, 75% A/25% B, 300 nL/min; 65 min, 65% A/35% B, 300 nL/min; 82 min, 10% A/90% B, 600 nL/min; 88 min, 10% A/90% B, 600 nL/min; 90 min, 96% A/4% B, 300 nL/min. The positive-ion mode with automated data-dependent MS/MS analysis was selected. Using full scans (350–1600 m/z) with FTMS at a mass resolution of 30,000, the ten most intense precursor ions were used for further analysis. The detailed parameters of MS acquisition settings were as follows: Analyzer, FTMS; Mass Range, Normal; Resolution, 30,000; Scan Type, Full; Polarity, Positive; Data Type, Profile; Act.Type, HCD; Iso.width (m/z),1.0; Normalized Collision Energy, 35.0; Act.Q, 0.25; Act.Times (ms), 10.00; First Mass (m/z), 350; Last Mass (m/z), 1600.

The raw MS files were analyzed by Proteome Discoverer 1.4 (Thermo Scientific). Raw data were searched against VdEno (VDAG_03029). Only PSMs with confidence at least medium and delta Cn better than 0.15 were considered. The peak filter was set to S/N Threshold (FT-only): 1.5. The trypsin and chymotrypsin as the protease with a maximum of two missed cleavages was allowed. The precursor mass tolerance was set to 20 p.p.m. and fragment mass tolerance was set to 0.05 Da. Carbamidomethyl (C) was set as a static modification, and SUMO1 (K), Oxidation (M), and Deamidated (N, Q) were set as variable modifications with a maximum of four modifications per peptide. The identified peptides with a minimal length of six amino acids and false discovery rate (FDR) = 0.01 as a cutoff were searched against the VdEno protein.

### Co-Immunoprecipitation and MS

Conidia and mycelia of *V. dahliae* were cultured in liquid Czapek-Dox medium for 3 days. The cultures were lysed with buffer (50 mM Tris-HCl pH 7.4, 150 mM NaCl, 0.5% NP-40, 20 mM N-ethylmaleimide (Sigma, E3876), 1× protease inhibitor) and centrifuged at 17949 g for 15 min at 4 °C. Eighty microliters of supernatant were used as the input, while the others were incubated with anti-Flag (Sigma, A2220) or anti-HA (AlpalifeB Inc, KTSM1305) agarose beads overnight at 4 °C. The input and IP samples were analyzed with a 1:5000 dilution of anti-HA (Easybio, BE2007-100), anti-Strep (Easybio, BE2038) and anti-Flag (F1804-200UG) antibody.

To identify the SUMO-specific protease target, total proteins extracted from the VdΔ*ulpb*/Strep-SUMO strain were enriched by Strep magnetic beads (*n* = 1). Protein A/G magnetic beads enrichment was used as a negative control (*n* = 1). Dried peptide samples were prepared as described above and resuspended in 0.1% formic acid (v/v). Peptide samples were subjected to EASY-nLC 1000 interfaced via a Nanospray Flex ion source to Orbitrap Fusion Tribrid mass spectrometer (Thermo Scientific). The trap column (Thermo Scientific Acclaim PepMap100, 100 μm × 2 cm, nanoViper C18) and analytical column (Thermo Scientific Easy Column, 10 cm long, 75 μm inner diameter, 3 μm resin) were used to load and separate peptides at a flow rate of 300 nL/min with a 60 min LC gradient composed of Solvent A (0.1% formic acid) and Solvent B (84% acetonitrile and 0.1% formic acid). The gradient was 0–60% buffer B for 50 min, 60–90% buffer B for 4 min, hold in 90% buffer B for 6 min. The mass spectrometer was operated in positive-ion mode, and MS data were acquired using a data-dependent top 20 method dynamically choosing the most abundant precursor ions from the survey scan (300–1800 m/z) for HCD fragmentation. Survey scans were acquired at a resolution of 70,000 at m/z 100 and resolution for HCD spectra was set to 17,500 at m/z 100.

MS/MS spectra were searched using MASCOT engine (Matrix Science, London, UK; version 2.2) against *Verticillium_dahliae* VdLs.17

ASM15067v2. The following options were used to identify the proteins. Peptide mass tolerance = 20 p.p.m., MS/MS tolerance = 0.1 Da, Enzyme = Trypsin, Missed cleavage = 2, Fixed modification: Carbamidomethyl (C), Variable modification: Oxidation(M), *P* value < 0.05 and FDR < 0.01. Peptides obtained from the Strep magnetic beads similar to those from A/G magnetic beads were removed to filter out contaminant proteins and nonspecific interactors. The rest of the peptides identified in Strep magnetic beads enrichment and the two-dimensional gel (Supplemental Methods) were selected for subsequent analysis.

### Enzyme activity assay

Conidia of *V. dahliae* from VdEno-GFP strains were cultured in liquid Czapek-Dox medium for 2 days, collected and incubated with the enzyme osmoticum, including zymolyase (MP, 08320921), lysing (Sigma, L1412) and driselase (Sigma, D8037) enzymes, for 2-3 h to prepare protoplasts. Protoplasts were lysed with extraction buffer (50 mM Tris pH 7.4, 150 mM NaCl, 1× Protease inhibitor) to obtain the cytoplasmic proteins. The pellet was lysed with buffer (50 mM Tris pH 7.4, 150 mM NaCl, 0.5% NP-40, 0.5% sodium deoxycholate, 1× Protease inhibitor) to obtain nuclear proteins. Cytoplasmic and nuclear VdEno were immunoprecipitated with anti-GFP agarose beads, and glycolytic enzyme activity was detected with enolase activity colorimetric assay kit (Biovision, K691-100). With a 1:5000 dilutions, Anti-histone 3 (EASYBIO, BE3222-100) and anti-α-tubulin antibody (EASYBIO, BE0031-100) were used to detect the nuclear and cytoplasmic markers.

### Data availability

All protein mass spectrometry raw data in this study are available via ProteomeXchange with identifiers PXD043396 (Strep magnetic beads), PXD043859 (Protein A/G magnetic beads), PXD043986 (protein spots from 2D gel) and PXD043398 (Identification of the SUMOylation sites). Source data are provided with this paper.

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

## Acknowledgements

The work was supported by the National Natural Science Foundation of China (No. 32020103003 and 32230003), Xinjiang Production and Construction Corps Science and Technology Plan Project (Grant No. 2022DB014), and the Strategic Priority Research Program of the Chinese Academy of Sciences (XDPB16). We thank Chuan-You Li for tomato seeds (MoneyMaker (LA2706) tomato plant), B. Thomma for the JR2 strain, and Jing-Fang Liu and Juan Li for MS data analysis.

## Author contributions

H.-S.G. and J.-H.Z. designed the experiments. X.-M.W., B.-S.Z., Y.-L.Z., and H.-W.W. conducted the experiments. H.-S.G., J.-H.Z., X.-M.W., B.-S.Z., Y.-L.Z., F.G., and J.Z. analyzed the data. H.-S.G., J.-H.Z., X.-M.W., B.-S.Z., Y.-L.Z. wrote and revised the manuscript.

## Competing interests

The authors declare that they have no conflict of interest.
