## [Peer Review File · Nature Communications]

REVIEWER COMMENTS

Reviewer #1 (Remarks to the Author):

In their manuscript, the authors investigate SUMOylation in the soil-borne fungus *Verticillium dahliae*. They show that the SUMO-specific protease (VdUlpB) mediates *V. dahliae* virulence, and the authors claim that this is due to deconjugating SUMO from enolase (VdEno). SUMOylated VdEno is mainly found in the nucleus where it represses expression of the VdSCP8 effector.

Overall, the manuscript has a logic flow, and presents novel and interesting work. The different sections of the work seem sound and robust, but I struggle making (and following) the connections. For example, between WT and VdUlpB knock-out the authors claim that only 31 proteins are differentially expressed. Firstly, if I stare at the gels, I see more proteins that seem differentially expressed, yet that have not been chosen by the authors. It feels weird nowadays that the “selection” is made by eye based on 2D gels, rather than based on a mass spectrometry analysis. Then, also it is unclear why only VdEno is chosen for functional analysis. Remarkably, it then. Also turns out that VdEno is fully responsible for the differential phenotype that was observed in the VdUlpB knock-out?

A similar issue I have with the “choice” of the VdSCP8 effector. This effector is “randomly” chosen (?) and then it turns out to be linked to VdEno?

Overall, I miss unbiased and genome-wide analyses that link the various phenomena in a non-biased manner rather than in the “random” manner in which the phenomena are presently linked.

The authors claim that VdEno acts as a glycolytic enzyme, which is demonstrated using an enolase activity colorimetric assay kit. However, it remains unclear what this actually means for the biology of *V. dahliae* infection. What does this enzyme activity do in fact? Does it contribute to virulence? What are the natural substrates?

Some of the claims cannot be assessed as the data are simply not presented. For example, the authors study VdEno SUMOylation. To this end, they co-express VdEno5K/5R-Flag and Strep-VdSUMO in an in vivo SUMOylation assay. For this they refer to Supplementary Fig. 4c, d, but this figure does not show such data, and also does not even have a panel d. Supplementary Figure 5 shows a similar experiment, but the constructs that are mentioned are not shown.

The whole manuscript needs to be carefully checked for language, grammar, and editing. For example, mention is made of “virulent effectors” (effectors are never virulent, the pathogen is),

there are many spelling mistakes, figure citations do not always seem to be correct/appropriate, “weird sentences” appear (e.g. lines 84-85: “In general, it should start with a brief introduction, followed by other sections.”).

Reviewer #2 (Remarks to the Author):

The manuscript ‘DeSUMOylation of enolase derepresses an effector 1 facilitating virulence in the soil-borne fungus *Verticillium dahliae*’ by Wu, et al, describes a novel virulence mechanism of the prominent crop pest, *Verticillium dahlia*. Due to the major economic impact of this fungus on commercially important crops, I believe that these findings can be suitable for publication in Nature Communications. A study of developing resistance to *Verticillium dahlia* was also recently published in the journal (Nat Commun. 2021 Nov 5;12(1):6426.) supporting the applicability of this current manuscript for Nature Communications. Overall, I believe that the experimental investigation described in this manuscript was carried out to a high standard. My main concern is the applicability of these results for other species infected by *Verticillium dahlia* and the potential for developing fungicides based on targeting this novel virulence mechanism.

Major comments:

1) These results are based on analysis in the cotton plant. A validation of the enolase nuclear translocation was also demonstrated using the leaves of *Nicotiana benthamiana*. To show the wider applicability of these results, I recommend that some of the major findings (leaf wilt, SUMOylation dependent prevention of nuclear translocation and repressor function) should be demonstrated in a second commercially important species, rather than *Nicotiana benthamiana*. This would validate the importance of enolase SUMOylation as a target for developing strategies to prevent *Verticillium dahlia* infection in agriculture.

2) The authors identify DeSUMOylation of enolase as a potential target for blocking *Verticillium dahlia* infection. A chemical inhibitor of SUMOylation is commercially available (Subasumstat (TAK-981)). I recommend that the authors use this inhibitor assess whether it is effective at preventing the enolase repressor function. This would provide additional evidence that DeSUMOylation prevents enolase nuclear translocation and support the development of chemical approaches to regulate SUMOylation.

3) I recommend that the Discussion section should be revised to cover the broader implications of enolase SUMOylation and nuclear translocation. In mammals, enolase acts as a transcriptional repressor via alternative splicing to produce the a truncated 37 kDa protein, MBP-1, which is primary localized in the nucleus (Front Cell Dev Biol. 2019 Apr 26;7:61). Although, both MBP-1 and ENO-1 may bind to target DNA sequences, only MBP-1 represses transcriptional activity. Can the authors comment on whether there are potential alternative splicing site(s) for *V. dahliae* enolase (VdEno)? Also, is it possible that SUMOylation could play a role in the transcriptional activity of mammalian enolase, where the transcription repressor is linked to tumor progression (via regulating c-myc expression)? How about the plant enolase(s): are they known to undergo nuclear localization or can they be modified by SUMOylation? I think it would be interesting if the authors provide their opinion on whether this SUMOylation mechanism is unique for VdEno, or may have a broader role in regulating the non-glycolytic activities of plant and animal enolases.

4) The authors state that 'The deletion of VdEno was not successful, indicating that VdEno's glycolysis activity is essential for *V. dahlia* development.' (Line 441) Was siRNA mediated knockdown attempted, as this could provide a partial reduction of VdENO expression?

Minor comments:

1) I recommend that the authors include a simplified diagram showing the major findings of their study. Due to the broad readership of Nature Communications, this would assist readers of the journal that are not familiar with plant science, but appreciate the wider implications of the authors' findings (e.g. include illustrations of the *Verticillium dahlia* fungus and infected plants in Figure 6F).

2) Line 84: this sentence seems to be related to manuscript formatting rather than the Introduction.

3) Could VdU1pB be a chemical target for preventing *Verticillium dahlia* infection? Are there 'druggable' sequence differences between VdU1pB and plant SUMO proteases?

4) Line 194: '1g' is in red font.

5) For the Figure 6F legend, I also recommend adding some explanation of the link between enhanced glycolysis and increased infection, for readers who are not expert in this research field.

Decision: Major revision

Reviewer #3 (Remarks to the Author):

The article is well written, the analyses are explained in detail, and the results are presented clearly.

The manuscript describes that VdUlpB is a SUMO protease and *V. dahliae* uses VdUlpB to deSUMOylate transcription repressor VdEno, which in turn derepresses a virulence effector gene VdSCP8 to promote plant infection; meanwhile, deSUMOylated VdEno regulates its cytoplasmic distribution to function as a glycolytic enzyme which essential for the fungal growth.

As a general comment, I think using the human SUMO1 antibody that recognises VdSUMO would strengthen the manuscript. In plants, AtSUMO1/2 are preferentially conjugated, while in humans is HuSUMO2/3. Since this information is not known for *V. dahlia*, the authors should check the accumulation of other SUMO proteins to put in context the general or specific effect of *V. dahliae* V592, since it had a dramatic effect in conjugating SUMO1.

Other comments:

Line 60. The authors briefly describe the SUMO pathway. However, I suggest providing the reader with more information about the different SUMO molecules and enzymes involved in plants but also in comparison with humans, since they used human SUMOylated targets and assume that HuSUMO 1 is orthologous to SUMO1 in plants.

Line 139. The nomenclature between the figure and the text is different, and therefore, it is difficult to follow.

Line 170. In figure Figure 2C, the authors show the accumulation of SUMOylated proteins in wild-type (WT) *V. dahliae* V592 and VdT-DNA, showing that in the absence of the VdUlpB, the level of SUMOylated proteins dramatically increases, suggesting VdUlpB has a major role in processing SUMO1/2. Which leads us to question if there are other Ulp/SENPs in *V. dahlia*? Does VdUlpB only recognise SUMO1?

Line 179. The authors use mass spectrometry to identify differences in protein abundance between V592 and VdT-DNA. By using 2D-electrophoresis, it is possible to visualise changes in protein size due to SUMOylation, but not with a total proteome. Therefore, is the author's objective to find downstream proteins potentially associated with transcription-dependent changes on SUMO, SUMO targets with different stability...? The objective did not come across, it should be explained in the text.

The straightforward approach would be doing a SUMO IP. Why did the authors not use this approach? By using this approach, the authors could identify novel targets of VdUlpB. The overlapping of the VdUlpB-dependent SUMO proteome and the total proteome (shown in this manuscript) would provide the identity of the pathways regulated by VdUlpB and a comprehensive view of the role of this protease, which seems to be critical for the regulation of SUMO targets in *V. dahlia*.

Response to comments

Reviewer #1 (Remarks to the Author):

In their manuscript, the authors investigate SUMOylation in the soil-borne fungus *Verticillium dahliae*. They show that the SUMO-specific protease (VdUlpB) mediates *V. dahliae* virulence, and the authors claim that this is due to deconjugating SUMO from enolase (VdEno). SUMOylated VdEno is mainly found in the nucleus where it represses expression of the VdSCP8 effector.

Overall, the manuscript has a logic flow, and presents novel and interesting work. The different sections of the work seem sound and robust, but I struggle making (and following) the connections. For example, between WT and VdUlpB knock-out the authors claim that only 31 proteins are differentially expressed. Firstly, if I stare at the gels, I see more proteins that seem differentially expressed, yet that have not been chosen by the authors. It feels weird nowadays that the “selection” is made by eye based on 2D gels, rather than based on a mass spectrometry analysis. Then, also it is unclear why only VdEno is chosen for functional analysis. Remarkably, it then. Also turns out that VdEno is fully responsible for the differential phenotype that was observed in the VdUlpB knock-out?

Response: Thank you for your positive recommendation on our manuscript.

SUMO modifications can change the molecular weight and/or the isoelectric point of target proteins, or cause degradation of target protein as well. Using 2D electrophoresis to separate proteins on the basis of their size and charge would therefore be a suitable approach to identify differentially modified proteins due to UlpB deficiency. The protein spots on the gels were quantified by using Image Master 2D Platinum software (Cytiva) as described in Supplemental Materials and Methods. The volume ratio between WT (V592) and mutant (Vd^{T-DNA}) higher than 2 and *P*-value < 0.05 as a cutoff, 31 spots were detected and proteins were identified by MS.

However, we agreed that this approach has a low sensitivity and throughput, which limited the number of proteins we could detect. To improve the identification of SUMO-modified proteins, we performed SUMO IP and mass spectrometry (MS) as Reviewer #3 suggested.

By using anti-SUMO1 (Proteintech, 10329-1-AP-100UL) for IP with Vd Δ ulpb

strain and anti-Strep for IP with Vd Δ ulpb/Strep-SUMO strain, MS analysis obtained 25 potential SUMOylated proteins in the ‘anti-SUMO1-IPed’-sample and 21 in the ‘anti-Strep-IPed’-sample (performed by APPLIED PROTEIN TECHNOLOGY). Only actin was found to coexist in both samples. Such few proteins were IPed and coexistent in both ‘anti-SUMO1-IPed’ and ‘anti-Strep-IPed’ samples, suggesting that the *in vivo* SUMO modification was a highly dynamic process.

Encouragingly, VdEno and actin were found to coexist in the 21 ‘anti-Strep-IPed’-proteins and the 31 ‘2D-gels-variant’-proteins.

Enolase was chosen as the target because we found that the VdUlpB mutant has a significant growth defect under cold stress, consistent with the findings that enolase is a cold stress response protein in *Arabidopsis thaliana* and *Botrytis cinerea*. Together with the identification of enolase in the new IP/MS assay, our data support enolase as a target of SUMOylation.

In this revision, the ‘anti-Strep-IPed’ data and more description have been added (Supplementary Table 1, lines 189-194, 197-201). With this additional data and description, we hope that the reviewer will agree that we have clarified the rationale for choosing VdEno for the primary candidate protein to investigate in more detail in this study, and our data provide substantiate the proposed link between VdUlpB and VdEno and *V. dahliae* virulence.

Undoubtedly, the growth and pathogenicity defects of the VdUlpB mutant were not attributed only to the dysregulation of VdUlpB-VdEno. However, VdUlpB-mediated deSUMOylation of VdEno, leading to distinct subcellular localizations and functions, certainly contributes to the fungal growth and pathogenicity.

A similar issue I have with the “choice” of the VdSCP8 effector. This effector is “randomly” chosen (?) and then it turns out to be linked to VdEno? Overall, I miss unbiased and genome-wide analyses that link the various phenomena in a non-biased manner rather than in the “random” manner in which the phenomena are presently linked.

Response: Thank you for pointing out this.

Due to unsuccessful ChIP-Seq and CUT&Tag (Kaya-Okur et al., 2019) thus far for this soil-borne pathogenic fungus (works by ourselves and Technology services companies: Gene Denovo Biotechnology Co. Guangzhou, China, <https://www.genedenovo.com/>), we were unable to perform genome-wide analysis of

the exact binding sequence in *V. dahliae*. (We would provide the reports of Gene Denovo Biotechnology Co. upon request.)

Therefore, we used the human *c-myc* DNA promoter element “TCGCGCTGAGTATAAAGCCGGTTT” bound by enolase to blast the genome sequence of VdLs.17 to search for putative VdEno targets. No conserved motif was detected; however, putative hits in promoters of a series of genes encoding secretory proteins were found (Supplemental Fig. 9a). A number of *V. dahliae* secretory effectors have been reported to modulate host immunity; thus, we examined the expression of the effector genes, and significantly reduced expression was detected for *VdSCP8* but not other *VdSCPs* in the *VdΔulpb* mutant compared to that in WT V592 (Supplemental Fig. 9b). Therefore, we reasonably chose *VdSCP8* for further study. (The related description is in lines 326-338).

The authors claim that VdEno acts as a glycolytic enzyme, which is demonstrated using an enolase activity colorimetric assay kit. However, it remains unclear what this actually means for the biology of *V. dahliae* infection. What does this enzyme activity do in fact? Does it contribute to virulence? What are the natural substrates?

Response: We thank the reviewer for this comment.

Enolase is a key enzyme in the glycolytic pathway that catalyzes the conversion of 2-phosphoglycerate to phosphoenolpyruvate (Wold and Ballou, 1957; Pancholi, 2001). The Enolase Activity Colorimetric/Fluorometric Assay Kit (Biovision, K691-100) was used to detect the intermediate product of the enolase-catalyzed reaction (Alam et al., 2020). Our results demonstrated that cytoplasmic VdEno has the capacity to catalyze the conversion of 2-phosphoglycerate to phosphoenolpyruvate. We have added these descriptions in this revision (lines 311-312, 315-317).

To determine whether the glycolytic activity of VdEno contributes to virulence, additional assays were performed and added.

First, we performed immunoprecipitation of VdEno by using anti-GFP with the cytoplasmic lysates from the VdEno-GFP and VdEno-GFP/*Δulpb* strains. A comparable glycolytic activity was detected with equal amounts of cytoplasmic IPed-VdEno proteins from each strain (added in Supplementary Fig. 12, lines 488-491), suggesting that the transcription factor activity of enolase, instead of glycolytic activity, contributes to *V. dahliae* virulence.

Furthermore, we generated an additional strain, *VdΔscp8/SCP8^{Pmut}*, in which the VdEno binding site in the promoter of *VdSCP8* was deleted. Increased expression of *VdSCP8* in *VdΔscp8/SCP8^{Pmut}* was detected compared with the complementary *VdΔscp8/SCP8* strain (added in Supplementary Fig 11c, lines 375-379). In agreement with the accumulation level of *VdSCP8*, *VdΔscp8/SCP8^{Pmut}* exhibited enhanced virulence in cotton plants compared with the *VdΔscp8/SCP8* strain (added in Supplementary Fig 11d, lines 391-394).

Taken together, our data support that SUMOylation of VdEno limits the expression of *VdSCP8*, rather than VdEno glycolytic activity per se, contributing to *V. dahliae* virulence.

All these additional data have been added in this revision.

Some of the claims cannot be assessed as the data are simply not presented. For example, the authors study VdEno SUMOylation. To this end, they co-express VdEno5K/5R-Flag and Strep-VdSUMO in an in vivo SUMOylation assay. For this they refer to

Supplementary Fig. 4c, d, but this figure does not show such data, and also does not even have a panel d. Supplementary Figure 5 shows a similar experiment, but the constructs that are mentioned are not shown.

Response: Thanks for your careful review.

We apologize for the missed labels and confusing sentences. We have now corrected “Supplementary Fig. 4c, d” to “Fig. 4c, d”. The construct information mentioned in Fig. 4d and Supplementary Figure 5d has been added in Supplemental Materials and Methods (supplemental information lines 50-51, 54-56).

The whole manuscript needs to be carefully checked for language, grammar, and editing. For example, mention is made of “virulent effectors” (effectors are never virulent, the pathogen is), there are many spelling mistakes, figure citations do not always seem to be correct/appropriate, “weird sentences” appear (e.g. lines 84-85: “In general, it should start with a brief introduction, followed by other sections.”).

Response: Thank you for pointing out these issues.

We have checked throughout the manuscript and the manuscript has been edited by AJE (American Journal Experts).

Reviewer #2 (Remarks to the Author):

The manuscript ‘DeSUMOylation of enolase derepresses an effector 1 facilitating virulence in the soil-borne fungus *Verticillium dahliae*’ by Wu, et al, describes a novel virulence mechanism of the prominent crop pest, *Verticillium dahlia*. Due to the major economic impact of this fungus on commercially important crops, I believe that these findings can be suitable for publication in Nature Communications. A study of developing resistance to *Verticillium dahlia* was also recently published in the journal (Nat Commun. 2021 Nov 5;12(1):6426.) supporting the applicability of this current manuscript for Nature Communications. Overall, I believe that the experimental investigation described in this manuscript was carried out to a high standard. My main concern is the applicability of these results for other species infected by *Verticillium dahlia* and the potential for developing fungicides based on targeting this novel virulence mechanism.

Response: Thanks for your positive recommendation and constructive suggestions on our manuscript.

Major comments:

1) These results are based on analysis in the cotton plant. A validation of the enolase nuclear translocation was also demonstrated using the leaves of *Nicotiana benthamiana*. To show the wider applicability of these results, I recommend that some of the major findings (leaf wilt, SUMOylation dependent prevention of nuclear translocation and repressor function) should be demonstrated in a second commercially important species, rather than *Nicotiana benthamiana*. This would validate the importance of enolase SUMOylation as a target for developing strategies to prevent *Verticillium dahlia* infection in agriculture.

Response: Thank you for your suggestion.

We have examined the UlpB effect on the pathogenicity for a tomato isolate VdJR2 strain during infection in tomato plants. UlpB-knockout mutant ($Vd\Delta ulpb^{JR2}$) showed no obviously developmental defect. However, $Vd\Delta ulpb^{JR2}$ strain showed significantly reduced degree of stunting of the plants on tomato when compared with the WT VdJR2 strain, which is in accord with that of the $Vd\Delta ulpb$ mutant of the cotton isolate V592. These data suggest that the UlpB effect on the pathogenicity was probably conserved in *V. dahliae*.

Regarding whether SUMOylation dependent prevention of nuclear translocation and repressor function of enolase took place in VdJR2 strain, it would take time for further study. Nevertheless, as a key glycolytic enzyme and a multifunctional protein, enolase is conserved in human, plant and fungus. Enolase and its posttranslational modifications, such as SUMOylation, might not be feasible targets for developing fungicides.

The assays with VdJR2 and mutant strains have been added in this revision (Supplementary Fig. 13, lines 498-501).

2) The authors identify DeSUMOylation of enolase as a potential target for blocking *Verticillium dahlia* infection. A chemical inhibitor of SUMOylation is commercially available (Subasumstat (TAK-981)). I recommend that the authors use this inhibitor to assess whether it is effective at preventing the enolase repressor function. This would provide additional evidence that DeSUMOylation prevents enolase nuclear translocation and support the development of chemical approaches to regulate SUMOylation.

Response: Thank you for your recommendation.

According to your suggestion, we incubated the WT V592 and *VdΔulpb* strains on TAK-981-containing medium. After 7 days of treatment, neither *VdΔulpb* nor V592 showed a significant difference in colony morphology compared to strains treated with DMSO (see below figures). Next, we examined the total protein SUMOylation level by western blotting with an anti-SUMO1 antibody (Proteintech, 10329-1-AP-100UL). No obvious alteration in protein SUMOylation level was detected in either strain with or without TAK-981 treatment (see below figures).

TAK-981 has been used in cell lines or animal models to inhibit SUMOylation (Langston et al., 2021; Rauth et al., 2021; Zhou et al., 2021; Jin et al., 2022). In our assays, TAK-981 was not successful in inhibiting SUMOylation in fungal cells. This may be due to the impediment of the fungal cell wall to TAK-981 permeating into and functioning in the fungal cells; alternatively, the enzyme molecules, such as E1, E2 or E3, in the SUMOylation pathway in fungal cells might differ from those in animal cells.

3) I recommend that the Discussion section should be revised to cover the broader implications of enolase SUMOylation and nuclear translocation. In mammals, enolase acts as a transcriptional repressor via alternative splicing to produce the a truncated 37

kDa protein, MBP-1, which is primary localized in the nucleus (Front Cell Dev Biol. 2019 Apr 26;7:61). Although, both MBP-1 and ENO-1 may bind to target DNA sequences, only MBP-1 represses transcriptional activity. Can the authors comment on whether there are potential alternative splicing site(s) for *V. dahliae* enolase (VdEno)? Also, is it possible that SUMOylation could play a role in the transcriptional activity of mammalian enolase, where the transcription repressor is linked to tumor progression (via regulating c-myc expression)? How about the plant enolase(s): are they known to undergo nuclear localization or can they be modified by SUMOylation? I think it would be interesting if the authors provide their opinion on whether this SUMOylation mechanism is unique for VdEno, or may have a broader role in regulating the non-glycolytic activities of plant and animal enolases.

Response: Thank you for your comments.

We have now discussed enolase function in depth in the revised version (lines 446-455, 463-467).

4) The authors state that ‘The deletion of VdEno was not successful, indicating that VdEno’s glycolysis activity is essential for *V. dahliae* development.’ (Line 441) Was siRNA mediated knockdown attempted, as this could provide a partial reduction of VdENO expression?

Response: Thank you for your suggestion.

We attempted to knockdown *VdEno* by transformation of a double-stranded RNA RNAi construct (dsVdEno) into V592. However, neither siRNAs nor reduced VdEno mRNA was detected in transformants.

We have previously reported that cotton plants export miRNAs to mediate target

cleavage in fungal cells (Zhang et al., 2016 Nature Plants) and a fungal miRNA mediates epigenetic repression of a fungal target gene (Jin et al., 2019 PTRS). The unsuccessful knockdown of target by the expression of dsRNA construct in this fungal cell might account for the more diverse and complicated fungal RNAi pathways than those in plants and animals (Jin et al., 2019; Zhao and Guo, 2022).

Minor comments:

1) I recommend that the authors include a simplified diagram showing the major findings of their study. Due to the broad readership of Nature Communications, this would assist readers of the journal that are not familiar with plant science, but appreciate the wider implications of the authors' findings (e.g. include illustrations of the *Verticillium dahlia* fungus and infected plants in Figure 6F).

Response: We have modified Figure 6F. Thanks.

2) Line 84: this sentence seems to be related to manuscript formatting rather than the Introduction.

Response: Deleted.

3) Could VdUlpB be a chemical target for preventing *Verticillium dahlia* infection? Are there 'druggable' sequence differences between VdUlpB and plant SUMO proteases?

Response: We compared the VdUlpB sequence with cotton *Gossypium hirsutum* SUMO-specific protease (GhUlp), and found that their sequences are not conserved (see below Figure). There are possible 'druggable' sequences between VdUlpB and GhUlp. However, as shown in Supplementary Fig. 1e and 1i, the VdUlpB, human SENP6 and SENP7 sequences are rather conserved. This might caution us that VdUlpB might not be a suitable target for developing fungicides.

```

1         10        20        30        40        50        60        70        89
VdUlpB  MGSKHKHVLKGTFFQPNNLKPEEGFMGRFRDVGHSVARFLPKFKDSDDDMG---AENAPFVNSTQKPHRSGLVLSQKATS--SR-PGDKRR
GhUlp   -----MDNIPCLVDADVGDGDCSCDVTPVASLGSDEDKDCILKEGNPKLNVSPESKSTHSEQQADLAKDSHEPRCTFTDMPRD
Consensus  L  DA          DV  VA          KD          G          APE  ST          A  L          S          D  R

90        100       110       120       130       140       150       160       178
VdUlpB  QSPEIDELAEDEKHHAK-RQ--R-RSNDLKTTPRVDIAALESRRTVAIAKRKDVVEVQRITDQLEGNRRSSRTYQQSSKQPFPAIDSDDE
GhUlp   SCLPEVPSFGRKRLNCALSNSPLNKFPVDMASDANESMAERTSP-SSDVAEDDVSLNGDMSDHCFG-DIPMDNLERTVVICIKYLKNPIQPE
Consensus  PEI  A          A          K  DL  S          IA  S  S  IA  DV          ISD  G  I          S          PD  I  D

179       190       200       210       220       230       240       250       267
VdUlpB  LNASPTKELDASGSSRPSTINGOVSPSNRKTPKGDIHHTKFKERQESQDEEVTLSRAVSGRVRFDAKCCGQTIRLRLNKKDTFTSLVPNK
GhUlp   YHTGVSVIFSPVKIEGSTVSEHQRTFSFESTIDDIISINCRWFPQR--VGYMTLKMKVLSKVVIAENAACDDEELKFTVIDPRWSEKHA
Consensus  S          SL          S  DI          K  Q          MTL  V  KV  DA  N  T

268       280       290       300       310       320       330       340       356
VdUlpB  AGEKLAMCWLTLTLRDRCHKIKFSKESQHLVPFR---SSTSITSPKLCIEIPSIS--DVIAITRWIFSAIATGAQDLKFKVKLEQPPR
GhUlp   AIMSLNFCYQALWNIMPDLIVEMDGGDSLVQRSYFPNFDEEEIYPKGDIAVSISKRDVDLLPETIVNDTIIDFYIKYLKNPIQPE
Consensus  A  L  QW  L          I          LV          I  PK  ID  SIS  DV  II  F          IKFLK  P  E

357       370       380       390       400       410       420       430       445
VdUlpB  TLQKVFENLYQSAPEQDRRRGLDRLCRQGRGSSPASPETAETTRRISLRDRMKVASNSPPDVKSQRRRSRS---NQLERRTDDM
GhUlp   RIRFHFNSERKLADLK---DESSISDGR---AAFLRVKTRKLDMFGDYIFIPVNFSLHWSLVICHFGAGFKDEDLKSKVP
Consensus  L  F  N  F          D  DK  DP          GR  A          TRKI  L  K  I          L  S  I  P  A          KSS

446       460       470       480       490       500       510       520       534
VdUlpB  ETLSAMNFFNNQWIKNYPDWDKIWEDKPLIYPASGKNRAQIIKDDIFRLEEHCLNDNLIVYLRYLQDLETENAGSERIIFMNPWF
GhUlp   CILHMDSIKGNHAGLKNLVQSYLWBEWKERKETSEDLSSKFLRVSLELPQQENSFDCLFLLYELFLAEAPPNFPFKITKFSN
Consensus  L          N          IWED  H  S  A          I          Q          L  F  L  YL  L  F  I  F  I  F

535       540       550       560       570       580       590       600       610       623
VdUlpB  YERLGQOKGRGVDYDAVKSWTAKIDLLSKDYIIVPVNEAAHWYLAIICHPGKLLPAVTDDKKTTVTNVDEEDMPOPESPIOSTSDPA
GhUlp   LNLDWLPTEASRTLQKLIFELLRSRSESSSDCSEPHSSRPPEEIGNGVEFVSKSVPEVACHGNPQASQGIEMILLASSSMRN
Consensus  A          K          IDI  SKD          E  H          G  L  S  S          S  I  SSS

624       630       640       650       660       670       680       690       700       712
VdUlpB  VDVQDSSETVRIVSDIASGKSPKIFKEITSGKEQTRKLDGG---KRPAPGDARIITLDSMGNTHSRTCNIKDYLVQEIKHKRQIDV
GhUlp   VETVNDSGLVREFFEPCVTAGSLLGQFQSFDQQPSYNGAVSPFEQEVQTQQFVYLASGETSFPQFGITSQACEVPYSSSGFVM
Consensus  VD  DS  LR  D  A          L  S  Q          GA  K  D  A          A  G  T          T  I          M

713       720       730       740       750       760       770       780       790       801
VdUlpB  ETPPRFGWTARGIPEQSDFSSCGIYLLAYVERFLKQPDQVISDIVYKKLDLWSDIDPVAMRGNIRELIIRLRQEQNVQRDKEKQAKQDA
GhUlp   GSSWNPGISGNKVDTSHETSCTSDDGDDIGIENNPIEDSVLVIEKRDTQEQSSVENVECLKKGPASEVLETSITEVPGASEDT
Consensus  S  G  SA  D  S  SSC          I  NP          II  KKD          D  V          IK  I          I  D  A  D

802       810       820       830       840       850       860       870       880       890
VdUlpB  KALKLAQKMDIFAVDSSPAHAAQTPGTTNRKPVQEIQVSSSDPPSSGKTGNFTAANPQDGRCPSNQLGSSPKPVRSPQSNQALIA
GhUlp   DKIHDKTGDADIPSKAHSSVVLHQNPGAAVNELDVDPVENMESQDNKT-----VSDQTLG---TVLNQLDRDSNLIE
Consensus  I          DIFA  S          Q  PG          V  I  S  S          S  Q  LG          V          LI

891       900       910       920       930       940       950       960       979
VdUlpB  LPRRSPKVDFLVDSSDIERLVSRHDNPLVPPILSSSSPEPGTPKRKEPHDELTSEADFPAPRRQFRNPKKRYDGVTAFKRSSWSYMSK
GhUlp   NEASCDEVQTIDDLPSKDNSMLLSNPSVAEALNQDSEVAENKEAIVNDSLAELSEOPAAKRMQLN---QSVEANKEAIVNDAEL
Consensus  V  IID          NP  M          E  A  K          D  L          P  A  R  Q  N          V  K  A          L

980       990       1000      1010      1020      1030      1040      1050      1068
VdUlpB  SGDHDERHESPEKQQTKHAERNRQAIVVVESPQSEKTSGKRTSPHFTLSPSVGRQSQRGLVTRNTPSRDDRGKMVVYGGTGRRAAK
GhUlp   SEQLAAKRMQLNQDSEVAE---NKEAIVNDSLAELS-----EQPAAKRMLSPSLEEVDS-----
Consensus  S          K  Q  S  AE  NK  AIV  D          A          LSPSL

1069       1086
VdUlpB  GDNNIDLTGADYSEYPYA
GhUlp   -----
Consensus

```

4) Line 194: '1g' is in red font.

Response: Corrected.

5) For the Figure 6F legend, I also recommend adding some explanation of the link between enhanced glycolysis and increased infection, for readers who are not expert in this research field.

Response: Added (lines 985-993). Thanks.

Decision: Major revision

Reviewer #3 (Remarks to the Author):

The article is well written, the analyses are explained in detail, and the results are

presented clearly. The manuscript describes that VdUlpB is a SUMO protease and *V. dahliae* uses VdUlpB to deSUMOylate transcription repressor VdEno, which in turn derepresses a virulence effector gene VdSCP8 to promote plant infection; meanwhile, deSUMOylated VdEno regulates its cytoplasmic distribution to function as a glycolytic enzyme which essential for the fungal growth.

As a general comment, I think using the human SUMO1 antibody that recognises VdSUMO would strengthen the manuscript. In plants, AtSUMO1/2 are preferentially conjugated, while in humans is HuSUMO2/3. Since this information is not known for *V. dahliae*, the authors should check the accumulation of other SUMO proteins to put in context the general or specific effect of *V. dahliae* V592, since it had a dramatic effect in conjugating SUMO1.

Response: Thanks for your positive recommendation and constructive suggestions on our manuscript.

SUMO modification in *V. dahliae* was detected with anti-(Hu)SUMO1 but not with anti-(Hu)SUMO2/3 in either WT V592 or V592/Strep-VdSUMO. As suggested, this result has now been added as a new Supplementary Figure 2c (line 169).

Furthermore, in *V. dahliae*, two SUMO homologs were found, but the transcript was detected only in one locus (Supplementary Fig. 2a, lines 141-142). Taken together, our data show that *V. dahliae* contains one functional SUMO protein that can be detected by the anti-(Hu)SUMO1 antibody.

Other comments:

Line 60. The authors briefly describe the SUMO pathway. However, I suggest providing the reader with more information about the different SUMO molecules and enzymes involved in plants but also in comparison with humans, since they used human SUMOylated targets and assume that HuSUMO 1 is orthologous to SUMO1 in plants.

Response: Thank you for your suggestion.

We have now added information about SUMO molecules and SUMO protease in humans, plants and yeast (lines 62-69, 73-75).

Line 139. The nomenclature between the figure and the text is different, and therefore, it is difficult to follow.

Response: Corrected.

Line 170. In figure Figure 2C, the authors show the accumulation of SUMOylated proteins in wild-type (WT) *V. dahliae* V592 and VdT-DNA, showing that in the absence of the VdUlpB, the level of SUMOylated proteins dramatically increases, suggesting VdUlpB has a major role in processing SUMO1/2. Which leads us to question if there are other Ulp/SENPs in *V. dahlia*? Does VdUlpB only recognise SUMO1?

Response: Thank you for this comment.

Two Ulp proteins in *V. dahliae*, VdUlpA and VdUlpB, have been identified. Phylogenetic analysis indicated that VdUlpA is classified into the yeast Ulp1 branch, while VdUlpB belongs to the yeast Ulp2 branch (Supplementary Fig. 1e).

As suggested, we created a *VdUlpA* mutant, *VdΔulpa*, to test whether VdUlpA regulated SUMOylation in *V. dahliae*. Whole proteins from the V592, *VdΔulpb* (as control) and *VdΔulpa* strains were extracted for Western blotting with an anti-human SUMO1 antibody. Enhanced protein SUMOylation was detected in *VdΔulpb* but not in *VdΔulpa* compared to V592 (lane 3). This result demonstrates that VdUlpA probably functions as a SUMO precursor-processing enzyme in *V. dahliae* similar to Ulp1 in yeast (Li and Hochstrasser, 2000), while VdUlpB mainly mediates the substrate for deSUMOylation in *V. dahliae*. We have added these new data to Fig 2d and lines 183-185.

As described above, two SUMO homologs were found in *V. dahliae*, but the

transcript was detected only in one locus and named VdSUMO. Therefore, we reason that VdUlpB recognizes VdSUMO.

Line 179. The authors use mass spectrometry to identify differences in protein abundance between V592 and VdT-DNA. By using 2D-electrophoresis, it is possible to visualise changes in protein size due to SUMOylation, but not with a total proteome. Therefore, is the author's objective to find downstream proteins potentially associated with transcription-dependent changes on SUMO, SUMO targets with different stability...? The objective did not come across, it should be explained in the text.

The straightforward approach would be doing a SUMO IP. Why did the authors not use this approach? By using this approach, the authors could identify novel targets of VdUlpB. The overlapping of the VdUlpB-dependent SUMO proteome and the total proteome (shown in this manuscript) would provide the identity of the pathways regulated by VdUlpB and a comprehensive view of the role of this protease, which seems to be critical for the regulation of SUMO targets in *V. dahlia*.

Response: Thank you for your comment and suggestion.

As described in the above response to reviewer #1, we agree that the 2D Electrophoresis has low sensitivity and throughput, which limits the number of proteins we can detect. According to your suggestion, we have performed SUMO IP and mass spectrometry (MS) to improve the identification of SUMO-modified proteins in this revision.

By using anti-SUMO1 (Proteintech, 10329-1-AP-100UL) for IP with the *VdΔulpb* strain and anti-Strep for IP with the *VdΔulpb*/Strep-SUMO strain, MS analysis (performed by APPLIED PROTEIN TECHNOLOGY) obtained 25 potential SUMOylated proteins in the 'anti-SUMO1-IPed'-sample (see below Table) and 21 in the 'anti-Strep-IPed'-sample (see Supplementary Table 1). Only actin and ADP/ATP carrier protein were found to coexist in both samples. Such few proteins were IPed and coexistent in both 'anti-SUMO1-IPed' and 'anti-Strep-IPed' samples, suggesting that the *in vivo* SUMO modification was a highly dynamic process.

Encouragingly, VdEno and actin were found to coexist in the 21 'anti-Strep-IPed'-proteins and the 31 '2D-gels-variant'-proteins.

Enolase was chosen as the target because we found that the VdUlpB mutant has a significant growth defect under cold stress, consistent with the findings that enolase is a cold stress response protein in *Arabidopsis thaliana* and *Botrytis cinerea*. Together

with the identification of enolase in this new IP/MS assay, our data support enolase as a target of SUMOylation.

In this revision, the ‘anti-Strep-IPed’ data and more description have been added (Supplementary Table 1, lines 189-194, 197-201). With this additional data and description, we hope that the reviewer will agree that we have clarified the rationale for choosing VdEno for the primary candidate protein to investigate in more detail in this study, and our data provide substantiate the proposed link between VdUlpB and VdEno and *V. dahliae* virulence.

Table. anti-SUMO1-IPed proteins.

No.	Accession	Score	Description
1	EGY14352.1	112	78 kDa glucose-regulated protein
2	EGY18111.1	77	actin
3	EGY21509.1	76	40S ribosomal protein S26E
4	EGY23050.1	64	glucose-6-phosphate 1-dehydrogenase
5	EGY19480.1	45	chitin binding protein
6	EGY16371.1	43	ADP,ATP carrier protein
7	EGY17352.1	43	40S ribosomal protein S11
8	EGY20379.1	42	hypothetical protein VDAG_10008
9	EGY13719.1	33	long-chain-fatty-acid-CoA ligase
10	EGY20340.1	30	retrograde regulation protein
11	EGY19567.1	26	hypothetical protein VDAG_09901
12	EGY16455.1	24	hypothetical protein VDAG_07619
13	EGY21133.1	23	hypothetical protein
14	EGY16307.1	22	cell division control protein
15	EGY13581.1	21	hypothetical protein VDAG_00263
16	EGY20889.1	21	(2R)-phospho-3-sulfolactate synthase
17	EGY21035.1	19	elongation factor 1-alpha
18	EGY22916.1	19	pre-mRNA-processing ATP-dependent RNA helicase PRP5
19	EGY13807.1	18	AFG1 protein
20	EGY19001.1	18	protein kinase domain-containing protein
21	EGY23278.1	16	hypothetical protein VDAG_04716
22	EGY16771.1	16	dihydroorotate dehydrogenase
23	EGY13931.1	15	THO complex subunit 2
24	EGY19725.1	15	hypothetical protein
25	EGY20814.1	14	rhamnogalacturonan lyase

- Alam, H., Tang, M., Maitituoheti, M., Dhar, S.S., Kumar, M., Han, C.Y., Ambati, C.R., Amin, S.B., Gu, B., Chen, T.Y., Lin, Y.H., Chen, J., Muller, F.L., Putluri, N., Flores, E.R., DeMayo, F.J., Baseler, L., Rai, K., and Lee, M.G. (2020). KMT2D Deficiency Impairs Super-Enhancers to Confer a Glycolytic Vulnerability in Lung Cancer. *Cancer cell* **37**, 599-617 e597.
- Jin, S., He, X., Ma, L., Zhuang, Z., Wang, Y., Lin, M., Cai, S., Wei, L., Wang, Z., Zhao, Z., Wu, Y., Sun, L., Li, C., Xie, W., Zhao, Y., Songyang, Z., Peng, K., Zhao, J., and Cui, J. (2022). Suppression of ACE2 SUMOylation protects against SARS-CoV-2 infection through TOLLIP-mediated selective autophagy. *Nature communications* **13**, 5204.
- Jin, Y., Zhao, J.H., Zhao, P., Zhang, T., Wang, S., and Guo, H.S. (2019). A fungal miRNA mediates epigenetic repression of a virulence gene in *Verticillium dahliae*. *Philos Trans R Soc Lond B Biol Sci* **374**, 20180309.
- Kaya-Okur, H.S., Wu, S.J., Codomo, C.A., Pledger, E.S., Bryson, T.D., Henikoff, J.G., Ahmad, K., and Henikoff, S. (2019). CUT&Tag for efficient epigenomic profiling of small samples and single cells. *Nature communications* **10**, 1930.
- Langston, S.P., Grossman, S., England, D., Afroze, R., Bence, N., Bowman, D., Bump, N., Chau, R., Chuang, B.C., Claiborne, C., Cohen, L., Connolly, K., Duffey, M., Durvasula, N., Freeze, S., Gallery, M., Galvin, K., Gaulin, J., Gershman, R., Greenspan, P., Grieves, J., Guo, J., Gulavita, N., Hailu, S., He, X., Hoar, K., Hu, Y., Hu, Z., Ito, M., Kim, M.S., Lane, S.W., Lok, D., Lublinsky, A., Mallender, W., McIntyre, C., Minissale, J., Mizutani, H., Mizutani, M., Molchinova, N., Ono, K., Patil, A., Qian, M., Riceberg, J., Shindi, V., Sintchak, M.D., Song, K., Soucy, T., Wang, Y., Xu, H., Yang, X., Zawadzka, A., Zhang, J., and Pulukuri, S.M. (2021). Discovery of TAK-981, a First-in-Class Inhibitor of SUMO-Activating Enzyme for the Treatment of Cancer. *Journal of medicinal chemistry* **64**, 2501-2520.
- Li, S.J., and Hochstrasser, M. (2000). The yeast ULP2 (SMT4) gene encodes a novel protease specific for the ubiquitin-like Smt3 protein. *Molecular and cellular biology* **20**, 2367-2377.
- Pancholi, V. (2001). Multifunctional alpha-enolase: its role in diseases. *Cellular and molecular life sciences : CMLS* **58**, 902-920.
- Rauth, S., Karmakar, S., Shah, A., Seshacharyulu, P., Nimmakayala, R.K., Ganguly, K., Bhatia, R., Muniyan, S., Kumar, S., Dutta, S., Lin, C., Datta, K., Batra, S.K., and Ponnusamy, M.P. (2021). SUMO Modification of PAF1/PD2 Enables PML Interaction and Promotes Radiation Resistance in Pancreatic Ductal Adenocarcinoma. *Molecular and cellular biology* **41**, e0013521.
- Wold, F., and Ballou, C.E. (1957). Studies on the Enzyme Enolase. *Journal of Biological Chemistry* **227**, 301-312.
- Zhao, J.H., and Guo, H.S. (2022). RNA silencing: From discovery and elucidation to application and perspectives. *J Integr Plant Biol* **64**, 476-498.
- Zhou, B., Zhu, Y., Xu, W., Zhou, Q., Tan, L., Zhu, L., Chen, H., Feng, L., Hou, T., Wang, X., Chen, D., and Jin, H. (2021). Hypoxia Stimulates SUMOylation-Dependent Stabilization of KDM5B. *Frontiers in cell and developmental biology*

9, 741736.

REVIEWERS' COMMENTS

Reviewer #2 (Remarks to the Author):

The authors have adequately responded to my comments. I recommend publication without further changes.

Reviewer #3 (Remarks to the Author):

The authors have adequately addressed my comments, and I believe the current version of the paper has improved significantly. I am happy to endorse the publication of the manuscript.